# A global monthly climatology of oceanic total dissolved inorganic carbon: a neural network approach

Daniel Broullón[1], Fiz F. Pérez[1], Antón Velo[1], Mario Hoppema[2], Are Olsen[3], Taro Takahashi[4,†], Robert M. Key[5], Toste Tanhua[6], Juana Magdalena Santana-Casiano[7] and Alex Kozyr[8]

[1]Instituto de Investigaciones Marinas, CSIC, Eduardo Cabello 6, 36208 Vigo, Spain

[2]Alfred Wegener Institute Helmholtz Centre for Polar and Marine Research, Postfach 120161, 27515 Bremerhaven, Germany

[3]Geophysical Institute, University of Bergen and Bjerknes Centre for Climate Research, Allégaten 70, 5007 Bergen, Norway

[4]Lamont-Doherty Earth Observatory of Columbia University, Palisades, NY 10964, USA

[5]Atmospheric and Oceanic Sciences, Princeton University, 300 Forrestal Road, Sayre Hall, Princeton, NJ 08544, USA

[6]GEOMAR Helmholtz Centre for Ocean Research Kiel, Düsternbrooker Weg 20D-24105 Kiel, Germany

[7]Instituto de Oceanografía y Cambio Global, IOCAG, Universidad de Las Palmas de Gran Canaria, Las Palmas de Gran Canaria, Spain

[8]NOAA National Centers for Environmental Information, 1315 East-West Hwy Silver Spring, MD 20910 USA

[†]Deceased

*Correspondence to*: Daniel Broullón (dbroullon@iim.csic.es)

**Abstract**

Anthropogenic emissions of $CO_2$ to the atmosphere have modified the carbon cycle for more than two centuries. As the ocean stores most of the carbon on our planet, there is an important task in unraveling the natural and anthropogenic processes that drive the carbon cycle at different spatial and temporal scales. We contribute to this by designing a global monthly climatology of total dissolved inorganic carbon ($TCO_2$) which offers a robust basis in carbon cycle modeling but also for other studies related to this cycle. A feedforward neural network (dubbed NNGv2LDEO) was configured to extract from the Global Ocean Data Analysis Project version 2.2019 (GLODAPv2.2019) and the Lamont-Doherty Earth Observatory (LDEO) datasets the relations between $TCO_2$ and a set of variables related to the former's variability. The global root-mean-squared error (RMSE) of mapping $TCO_2$ is relatively low for the two datasets (GLODAPv2.2019: 7.2 µmol kg$^{-1}$; LDEO: 11.4 µmol kg$^{-1}$) and also for independent data, suggesting that the network does not overfit possible errors in data. The ability of NNGv2LDEO in capturing the monthly

variability of $TCO_2$ was testified through the good reproduction of the seasonal cycle in ten time-series stations spread over different regions of the ocean (RMSE: 3.6 to 13.2 µmol kg$^{-1}$). The climatology was obtained by passing through NNGv2LDEO the monthly climatological fields of temperature, salinity and oxygen from World Ocean Atlas 2013, and phosphate, nitrate and silicate computed from a neural network fed with the previous fields. The resolution is 1° x 1° in the horizontal, 102 depth levels (0-5500m) and monthly (0-1500 m) to annual (1550-5500 m), and it is centered in the year 1995. The uncertainty of the climatology is low when compared with climatological values derived from measured $TCO_2$ in the largest time-series stations. Furthermore, a computed climatology of partial pressure of $CO_2$ (pCO$_2$) from a previous climatology of total alkalinity and the present one of $TCO_2$ supports the robustness of this product through the good correlation with a widely used pCO$_2$ climatology (Landschützer et al., 2017). Our $TCO_2$ climatology is distributed through the data repository of the Spanish National Research Council (CSIC; http://dx.doi.org/10.20350/digitalCSIC/10551, Broullón et al., 2020).

## 1 Introduction

The ocean is the major carbon reservoir of the Earth. Most of this carbon occurs as dissolved inorganic carbon ($TCO_2$, also known as DIC or $C_T$) (Ciais et al., 2013; Tanhua et al., 2013). Three species make up $TCO_2$: dissolved $CO_2$ (generally considered as the sum of the dissolved $CO_2$ itself ($CO_2(aq)$) and carbonic acid ($H_2CO_3$)), bicarbonate ion ($HCO_3^-$) and carbonate ion ($CO_3^{2-}$). The relative concentrations of these species with respect to each other determine the seawater pH (Zeebe and Wolf-Gladrow, 2001). The seawater $CO_2$ chemistry system can be represented as a set of chemical equilibria reactions that describes the speciation of the various ions of $TCO_2$ as follows:

$$CO_2(g) \rightleftharpoons CO_2(aq)$$

$$CO_2(aq) + H_2O \rightleftharpoons H_2CO_3$$

$$H_2CO_3 \rightleftharpoons H^+ + HCO_3^-$$

$$HCO_3^- \rightleftharpoons H^+ + CO_3^{2-}$$

Since the Industrial Revolution, the concentration of $TCO_2$ in the global ocean has increased, generally to a certain depth level (depending on the particular processes in each ocean area) due to the entry of $CO_2$ into the seawater from the atmosphere (Sarmiento and Gruber, 2002; Doney et al., 2009; Vázquez-Rodríguez et al., 2009; Bates et al., 2012; Sallée et al., 2012; Khatiwala et al., 2013). The uptake is driven by the increasing partial pressure of $CO_2$ (pCO$_2$) in the atmosphere relative to the ocean, generated by the anthropogenic emissions of $CO_2$ that cause an annual net flux of this gas into the ocean (Le Quéré et al., 2018). Accompanying the change in $TCO_2$, the pH and carbonate ion concentration have been declining because of the anthropogenic process previously mentioned, these changes being reflected in the proportions of the chemical species of $TCO_2$ (Kleypas and Langdon, 2000; Orr et al., 2005). These changes in seawater chemistry framed in the ocean acidification process can negatively influence various processes

involving marine organisms such as calcification, growth and survival (Orr et al., 2005; Fabry et al., 2008; Hendriks et al., 2010; Hoegh-Guldberg and Bruno, 2010; Kroeker et al., 2013).

In addition to the secular trends driven by the uptake of anthropogenic $CO_2$, ocean $TCO_2$ varies both temporally and spatially as a consequence of several natural processes. This variability may reach values of 15% of the mean $TCO_2$ value in the ocean (Lee et al., 2000). The processes that increase $TCO_2$ are: net flux of $CO_2$ from the atmosphere to the ocean, organic matter remineralization and the dissolution of calcium carbonate ($CaCO_3$). The processes that reduce $TCO_2$ are: net flux of $CO_2$ from the ocean to the

atmosphere, primary production and calcification. Advection and mixing also influence the variability of $TCO_2$ in these two ways (Sabine et al., 2002). In the surface ocean, the main variables influencing the variability of $TCO_2$ are temperature and salinity (Weiss et al., 1982; Lee et al., 2000; Wu et al., 2019) through the modification of the solubility of $CO_2$, affecting the seawater $pCO_2$ (which is almost instantaneous) and thus the air-sea $CO_2$ flux, which eventually drives the change in $TCO_2$ over time..

Nutrients and oxygen can also reflect the processes that modify the concentration of $TCO_2$ through their consumption and release, like during the cycling of organic matter (Körtzinger et al., 2001; Bauer et al., 2013). From products generated with measured data (Key et al., 2004; Takahashi et al., 2014; Lauvset et al., 2016) and in modeling studies (e.g., Doi et al., 2015), it is known that the global surface distribution of $TCO_2$ follows a zonal gradient: there is a reduction of its concentration from the poles to the equator

reflecting the processes that control its variability. Key et al. (2004) emphasize that this distribution is associated to the distribution pattern of nutrients. Recently, Wu et al. (2019) found that the distribution of surface salinity-normalized $TCO_2$ (nDIC) has two main drivers: temperature and upwelling. At depth, the variation shown in almost any measured profile of $TCO_2$ mainly reflects the remineralization of organic matter and, to a lesser extent, the dissolution of $CaCO_3$ (Millero, 2007), resulting in an increase in $TCO_2$

from the surface to the intermediate depths.

Understanding the distribution and variability of $TCO_2$ in the ocean and its secular trends driven by anthropogenic carbon uptake is needed to assess the magnitude and possible impacts of ocean acidification. It is also necessary for the evaluation of numerical models that include the carbon cycle and their estimates of past, current and future ocean carbon cycle behavior (e.g., Yool et al., 2013; Aumont et al., 2015;

Butenschön et al., 2016; Le Quéré et al., 2016; Goris et al., 2018). Seasonality of $TCO_2$ and the horizontal and vertical variability underscore the necessity to design a climatology with both monthly and spatial resolutions according to the processes that influence this variable on a global scale. The existing climatologies of $TCO_2$ do not include all these characteristics collected together. Key et al. (2004) and Lauvset et al. (2016) built an annual climatology in 33 depth levels using interpolation techniques over data

from Global Ocean Data Analysis Project version 1 (GLODAPv1; Key et al., 2004) and GLODAPv2 (Key et al., 2015; Olsen et al., 2016), respectively. Takahashi et al. (2014) published a monthly climatology for the surface ocean computed from climatologies of $pCO_2$ and total alkalinity ($A_T$). Other studies used the co-variability between $TCO_2$ and other more commonly measured variables discussed above for mapping/gap-filling via empirical regressions and neural networks. Lee et al. (2000) used temperature and

nitrate to compute surface nDIC with an area-weighted error of $\pm 7$ $\mu mol$ $kg^{-1}$. Sauzède et al. (2017) and Bittig et al. (2018) trained neural networks with GLODAPv2 data to compute $TCO_2$ over the depth range

0-8000 m with an accuracy of $\pm 9$ µmol kg$^{-1}$ and $\pm 7.1$ µmol kg$^{-1}$, respectively. The input variables used in those studies were location, pressure, temperature, salinity, dissolved oxygen and time.

In the present study, we introduce the use of neural networks for going one step further in the design of a

climatology. We have generated a climatology of $TCO_2$ with a resolution consistent with that of the climatology of $A_T$ of Broullón et al. (2019): horizontal resolution of 1°x1°, 102 depth levels between 0 and 5500 m and a monthly (0-1500 m) and annual (1550-5500 m) temporal resolution. The availability of global databases containing variables of the seawater $CO_2$ system with more and more data (e.g., GLODAPv2.2019, Lamont-Doherty Earth Observatory database (LDEO; Takahashi et al., 2017), Surface

Ocean $CO_2$ Atlas (SOCAT; Bakker et al., 2016)) and the great ability of the neural networks to interpolate as shown in other climatological studies about $CO_2$ system variables (Landschützer et al., 2014; Broullón et al., 2019), show the appropriateness of this approach for generating a global monthly climatology covering more than the surface ocean.

**2 Methodology**

**2.1 Neural network design**

A feed-forward neural network was configured to compute $TCO_2$ in the global ocean and to create a global climatology based on the good results previously obtained with this method in similar studies (e.g., Broullón et al., 2019). Briefly, a neural network of this type (Fig. S1a) is used to extract relationships between a set of input variables and a target one through a training process. At this stage, the inputs are passed through

different parallel layers composed by a tunable number of neurons to reach values as closest as possible to the target ones (Fig. S1a). Initially, all inputs enter in each neuron of the first layer where they are being multiplied by different weights depending on the neuron they go. Inside the neurons (Fig. S1b), the results of the previous operation are summed and a bias is added. The obtained value inside each neuron is passed through an activation function which yields an output. The outputs of each neuron in each layer go to the

following layer suffering the same process described to this point. In the last layer, which is composed by one neuron, a unique value for the target variable is calculated for each pair of inputs-target. This value is compared to the desired one and the difference between both values is backpropagated through the entire network in order to adjust the weights and biases, and to start again the processes and reach an accurate output value after multiple iterations. A complete description of the most common algorithms used to

backpropagate and minimize the errors can be founded in Rumelhart et al. (1986), Levenberg (1944) and Marquardt (1963).

The method used here is equivalent to that fully described by Broullón et al. (2019) for $A_T$. In addition to the target variable ($TCO_2$ instead of $A_T$), the main changes in the present study compared to that of Broullón et al (2019) are the inclusion of the input variable "year", accounting for the anthropogenic increase of the

$TCO_2$ pool, and the use of the $pCO_2$ database from LDEO (Takahashi et al., 2017) in addition to the extended GLODAPv2.2019 (Olsen et al. 2019) to enable more robust $TCO_2$ estimates in the surface ocean. Similar to Broullón et al. (2019), the neural networks were trained using the Levenberg-Marquardt method (Levenberg, 1944; Marquardt, 1963) through the *trainlm* function (detailed in Beale et al., 2018) in

MATLAB. The splitting of the database used in the present study (see Sect. 2.2) in the sets needed for training and testing the network is depicted in Fig. 1. The data were randomly associated to each dataset to capture (training) and evaluate (test) all possible variability. The input variables are temperature, salinity, phosphate, nitrate, silicate, oxygen, sample position and year (Fig. S1a). The number of neurons tested in the unique hidden layer to find the best neural network was 16, 32, 64, 128 and 256. Ten networks were trained for each number of neurons. The criteria to select the final number of neurons are based on a trade-off between the root-mean-squared error (RMSE — between the measured $TCO_2$ and that estimated by the neural network) on the one hand, and the generalization of the network (to prevent overfitting, maintaining a similar error in the training and in the test sets) on the other hand. Furthermore, an additional criterion based on the influence of each input variable on the $TCO_2$ extracted with the connection weight approach (Olden and Jackson, 2002) was followed to ensure that biogeochemical input variables have a larger influence on the $TCO_2$ estimates than the input variables related to sample position for selecting a proper network. The influence of each input variable on the computed $TCO_2$ was obtained from Eq. (1):

$$C_i = \sum_{k=1}^{H} w_{ik} \cdot w_k \tag{1}$$

where $C_i$ is the relative importance of the input variable i, H the number of neurons in the hidden layer, $w_{ik}$ the weight of the connection between the variable i and the neuron k of the hidden layer, and $w_k$ is the weight of the connection between the neuron k of the hidden layer and output layer.

## 2.2 Data

We included the LDEO database version 2016 (Takahashi et al., 2017; https://www.nodc.noaa.gov/ocads/oceans/LDEO_Underway_Database, last access: 13 November 2017), because it contains significantly more data in the surface layer than GLODAPv2.2019. Since the higher variability in the surface layer may lead to high errors in modeling variables of the seawater $CO_2$ system (e.g., Carter et al., 2018; Bittig et al., 2018; Broullón et al., 2019), including the LDEO database should force the network to reach a more robust fit. The idea is that these additional data probably have more different relationships between input variables and $TCO_2$ to help the neural network to adequately capture spatiotemporal variability. The $pCO_2$, temperature and salinity data from LDEO were monthly-averaged for each year in a 1°x1° grid. The points where the standard deviation of the averaged $pCO_2$, temperature and salinity were greater than $\pm20$ µatm, 1.5°C and 0.5, respectively, were discarded, since the objective is to capture the monthly variability and therefore an extremely high sub-monthly variability could lead to errors. To obtain $TCO_2$ values from the LDEO data, an additional variable of the $CO_2$ system is necessary, for which we take $A_T$ computed using the neural network NNGv2 of Broullón et al. (2019). The input variables required by NNGv2 were obtained from: 1) temperature and salinity from LDEO; 2) filtered oxygen from World Ocean Atlas version 2013 (WOA13; see Broullón et al., 2019); 3) phosphate, nitrate and silicate computed with CANYON-B (Bittig et al., 2018) using the previous variables as inputs. Finally, $TCO_2$ was calculated from this $A_T$ and the averaged $pCO_2$ using the MATLAB-version of the $CO_2SYS$ program (van Heuven et al., 2011); we used the dissociation constants of Mehrbach et al. (1973) (as refit by Dickson and Millero, 1987) and the borate dissociation constant of Dickson (1990). Note that we used

this software and set of constants for all seawater $CO_2$ chemistry calculations in the present study. The thus calculated $TCO_2$ and the associated input variables were used as a part of the training and testing data for the neural networks created here. The final number of data points derived from LDEO was 54572.

To represent interior ocean conditions, the GLODAPv2.2019 database (Olsen et al., 2019) was added to the LDEO dataset for training and testing the neural network. Only samples which had data for all input variables and $TCO_2$ were used. This database was included in two ways: 1) Only samples where all variables passed the $2^{nd}$ quality control (n=287953) (Olsen et al., 2016; Olsen et al., 2019; hereafter abbreviated Gv2QC) and 2) all samples (n=321647) (hereafter abbreviated Gv2). Therefore, two neural networks options were trained and tested: NNGv2QCLDEO and NNGv2LDEO, respectively.

## 2.3 Comparison of methods

We compared our method with CANYON-B of Bittig et al. (2018), where also $TCO_2$ values were computed from multiple input variables. Both methods are based on neural networks but with certain differences as summarized in Table 1.

An error analysis was carried out in the same areas for which this was done by Broullón et al. (2019) for $A_T$ and in several depth ranges (0-50 m, 50-200 m, 200-500 m, 500-1000 m and 1000 m-bottom) for the two methods (our method and CANYON-B) and for the two datasets (Gv2QC and LDEO). The Gv2QC database was analyzed in this section instead of Gv2 because in the designing of CANYON-B only quality-controlled data were included. The analysis of CANYON-B using the LDEO dataset is useful to evaluate the validity of the approach followed by converting $pCO_2$ to $TCO_2$ since CANYON-B has not been trained with this dataset.

Computed $pCO_2$ from $A_T$ and $TCO_2$ derived from neural networks was also evaluated in the LDEO dataset to assess the adequacy of including this dataset in our approach and to assess the ability of NNGv2 of Broullón et al. (2019) and the present $TCO_2$ neural network to compute other variables of the seawater $CO_2$ system. Furthermore, we compared the magnitude of the errors with the ones obtained by Landschützer et al. (2014), in which $pCO_2$ is computed directly with a neural network, to evaluate the accuracy of our computed $pCO_2$.

## 2.4 Validation

In addition to the ability of computing $TCO_2$ using the Gv2 and LDEO test sets, the neural network has been tested using independent data from ten ocean time series, located in different regions of the world ocean (data were obtained from https://www.nodc.noaa.gov/ocads/oceans/time_series_moorings.html, last access: 4 June 2019): Hawaii Ocean Time-series (HOT ALOHA and HOT ALOHA SURFACE; Dore et al., 2009), Bermuda Atlantic Time-series Study (BATS; Bates et al., 2012), European Station for Time-series in the Ocean at the Canary Islands (ESTOC; González-Dávila et al., 2010), Iceland Sea Time-series (ICELAND; Olafsson et al., 2010) Irminger Sea Time-series (IRMINGER; Olafsson et al., 2010), Kyodo North Pacific Ocean Time-series (KNOT; Wakita et al., 2010), K2 (Wakita et al., 2010), Ocean Weather Station Mike (OWS; Gislefoss et al., 1998) and Kerguelen Islands in the Indian sector of the Southern

Ocean (KERFIX; Jeandel et al., 1998). CANYON-B was also used to compute $TCO_2$ in the time series to show the differences between that method and ours. The $TCO_2$ values were obtained by feeding the neural networks with the measured values of the input variables at each time series. The data from these time series allow us to test the ability of the neural network to reconstruct not only the seasonal variability of $TCO_2$ at the various locations and depths sampled, but also its long-term trends. For the trend analyses, the measured and estimated $TCO_2$ values were deseasonalized following Bates et al. (2014).

As an additional test, the measured $pCO_2$ or the $pCO_2$ calculated from measured $TCO_2$ and $A_T$ at the time series stations were compared with $pCO_2$ calculated from the neural network generated values of $A_T$ and $TCO_2$. This provides insight in the combined performance of the NNGv2 of Broullón et al. (2019) and the neural network designed in the present study. Furthermore, we compared the magnitude of the errors to that obtained by Landschützer et al. (2014) for some of the time series.

### 2.5 Climatology of $TCO_2$

We used the selected network, based on the results of the analyses described above, to construct a climatology of $TCO_2$. Climatologies of the input variables were passed through the network to obtain the climatological fields of $TCO_2$. The spatiotemporal resolution of the product is determined by that of the climatologies used as inputs: 1° x 1° horizontal resolution, 102 upper depth levels of the WOA13 and monthly (for 0-1500 m depth) to annual (for 1550-5500 m depth) temporal resolution. Temperature and salinity climatologies were obtained from WOA13 objectively analyzed fields (Locarnini et al., 2013; Zweng et al., 2013; https://www.nodc.noaa.gov/OC5/woa13/woa13data.html, last access: 6 February 2017). Oxygen, phosphate, nitrate and silicate climatologies were taken from Broullón et al. (2019) (http://dx.doi.org/10.20350/digitalCSIC/8644, last access: 1 August 2019). These climatologies of nutrients were created using the objectively analyzed climatologies of temperature, salinity and oxygen (Garcia et al., 2014 filtered, see Broullón et al. (2019)) from WOA13 in CANYON-B (Bittig et al., 2018). As a year input is needed, we decided to center the $TCO_2$ climatology in 1995 based on the time distribution of the data used to create the WOA13 climatologies: World Ocean Database 2013 (Boyer et al., 2013).

The computed climatological values were compared with those from measured data to assess the uncertainty of the climatology, since WOA13 does not offer an uncertainty field with the objectively analyzed climatologies. Unfortunately, only two locations have enough measured data to calculate a pure climatological value of $TCO_2$ for each month: HOT ALOHA and BATS. The measured values were monthly averaged at several depth levels and the anthropogenic carbon as calculated by Lauvset et al. (2016) was added or subtracted to correct the data to the reference year of the climatology according to:

$$TCO_2^{year_2} = TCO_2^{year_1} - C_{ant_{2002}}[(1 + 0.0191)^{(year_1 - 2002)} - (1 + 0.0191)^{year_2 - 2002}] \qquad (2)$$

where $TCO_2^{year_2}$ is the $TCO_2$ corrected to $year_2$, which is the reference year of the climatology, $TCO_2^{year_1}$ is the $TCO_2$ measured in $year_1$, $C_{ant_{2002}}$ is the anthropogenic carbon for 2002 and 0.0191 is the annual increase rate derived from the scaling factor determined by Gruber et al. (2019) for the global ocean between 1994 and 2007.

We compared our climatology with previously published climatologies of $TCO_2$. The surface monthly climatology created by Takahashi et al. (2014) was used to assess the spatiotemporal differences in the surface layer. The annual climatology of Lauvset et al. (2016) was used to evaluate the spatial differences in the deeper parts of the ocean. For the comparisons, the climatologies of Takahashi et al. (2014) and Lauvset et al. (2016) were adjusted to the year 1995 subtracting the anthropogenic carbon ($C_{ant}$) of Lauvset et al. (2016) as in the Eq. (2).

Finally, a surface climatology of $pCO_2$ was computed from the $TCO_2$ climatology of the present study and the $A_T$ climatology of Broullón et al. (2019) to assess the potential of computing climatologies of other variables of the seawater $CO_2$ system. For comparison, the updated monthly $pCO_2$ climatology from Landschützer et al. (2016) (Landschützer et al., 2017, last access: 30 July 2019) was used. The values between 1981 and 2010 were averaged to obtain the climatological year 1995. The variable selected from Landschützer et al. (2017) was that labeled as spco2_raw (sea surface $pCO_2$) in the netCDF file.

It should be noted that the RMSE and the bias were obtained for all the comparisons, the last statistic being computed as the difference between the measured (or computed by the method to compare) $TCO_2$ and the one obtained with the neural network of the present study.

## 3. Results

### 3.1 Neural network analysis

Following the established criteria to obtain the optimal number of neurons, the configuration with 128 neurons in the hidden layer was selected. From the ten networks trained with this number of neurons for each approach (NNGv2LDEO and NNGv2QCLDEO), the ones with the lowest influence of the position input variables were selected. These two networks present a similar RMSE in both training and test datasets, showing there is no overfitting. Because in Gv2QC both NNGv2LDEO and NNGv2QCLDEO produce the same global RMSE (6.1 µmol kg$^{-1}$), it is likely that the Gv2 dataset contains high-quality measurements and the possible errors on the non-QC data of this dataset are clearly avoided by the network; otherwise NNGv2LDEO should have a higher RMSE in the test dataset than NNGv2QCLDEO because of an overfitting of the errors in the Gv2 dataset. The same holds for the LDEO dataset. The network properly fitted $TCO_2$ derived from LDEO, since it does not significantly increase the global RMSE relative to a network only trained with Gv2. Therefore, we decided to continue with NNGv2LDEO only since it has fitted more relationships between variables (e.g., Gv2 has more data points than Gv2QC in the Mediterranean Sea) providing a more robust fitting. For this network, the influence of each input variable on the computed $TCO_2$ is depicted in Fig. S2. The position variables together (latitude, clongitude, slongitude and depth) have no more than 30% influence, allowing biogeochemical variables to be the main ones responsible for the variability of $TCO_2$. Furthermore, the input variable year has an influence lower than 5%. This is probably responsible for capturing the positive interannual trend due to the $TCO_2$ increase derived from anthropogenic emissions of $CO_2$ to the atmosphere (see Sect. 3.2).

The global RMSE is quite low for the Gv2 dataset and for the LDEO dataset (Fig. 2). The measured and the computed data are highly correlated (Fig. 2) and the bias is negligible in both datasets. The higher RMSE in the LDEO dataset likely results from the higher variability of $TCO_2$ in the surface layer and from uncertainties in its calculation from $pCO_2$.

The RMSE by area and depth for NNGv2LDEO and CANYON-B in Gv2QC is shown in Table 2. The highest errors for the two methods are in the 0-50 m layer for the Gv2QC dataset and the LDEO dataset. These errors get smaller with increasing depth for all areas and the depth-weighted RMSE of the two methods is not significantly different below 50 m. In the LDEO dataset, NNGv2LDEO produces a lower error than CANYON-B, except for two areas: East GIN (Greenland, Iceland and Norwegian) Seas and the Bengal Basin (Table 2), although there are only 9 and 13 data points, respectively, in each area. Interestingly, CANYON-B is able to reproduce the $TCO_2$ data derived from the complete LDEO dataset with a lower error than the one it obtains for the complete Gv2QC dataset in the surface ocean (RMSE LDEO: 16.4 μmol kg$^{-1}$; RMSE Gv2QC (0-5 m): 17.8 μmol kg$^{-1}$), supporting the approach of computing reliable $TCO_2$ values from the $pCO_2$ of LDEO and the $A_T$ computed with NNGv2 (Broullón et al., 2019), since CANYON-B was not trained with the LDEO database. A similar result was obtained for NNGv2LDEO but with a higher difference between the two errors (RMSE LDEO: 11.4 μmol kg$^{-1}$; RMSE Gv2QC (0-5m): 17.1 μmol kg$^{-1}$). Finally, the surface RMSE towards LDEO data of NNGv2LDEO is clearly lower than that of CANYON-B. This shows the value of including $pCO_2$ derived surface $TCO_2$ among the training data, through which there are more fitted relations in our new method.

For data from Gv2 where no QC was performed for at least one of the variables used in the present study (Gv2noQC), the RMSE also decreases with increasing depth: <50 m: 22.5 μmol kg$^{-1}$; 50-200 m: 9.8 μmol kg$^{-1}$; 200-500 m: 7 μmol kg$^{-1}$; 500-1000 m: 5.4 μmol kg$^{-1}$; >1000 m: 5.4 μmol kg$^{-1}$. Thus, the error in Gv2noQC is similar to that in the areas with the highest error in Gv2QC (Table 2; except in Beaufort Sea, where the error is considerably higher). However, the higher error in Gv2noQC is mainly caused by the samples located in the Arctic Ocean, since cruises in the Atlantic and Pacific oceans are modeled with a very low error. Therefore, using Gv2noQC does not imply the introduction of low-quality data in our study, otherwise the network would not compute $TCO_2$ with low errors in Gv2QC because of an overfitting of the possible low-quality data that Gv2noQC could contain.

In general, the highest differences between measured and estimated $TCO_2$ occurs in the high latitude surface oceans (Figs. 3 and 4). In Gv2, 40% of the samples with differences beyond ±3RMSE (3 times RMSE; threshold selected to refer samples with large residuals) are in latitudes greater than 70º N. In the LDEO dataset, 39% of the samples with differences beyond ±3RMSE are from latitudes south of 70º S. These samples where RMSE is high are 7.5% of the total north of 70º N in Gv2 and 42% of the total south of 70º S in LDEO. The samples with low salinities have the highest errors (Fig. 4). 41.5% of the samples in Gv2 and 43% in LDEO with differences beyond ±3RMSE have salinities below 33. Furthermore, in the LDEO dataset, the number of samples with residuals beyond ±3RMSE increases with increasing standard deviation of both $pCO_2$ and salinity in the monthly averaging in each pixel in the LDEO subset (Fig. S3). This result shows the difficulty of modeling areas with a high sub-monthly variability in $pCO_2$ and salinity and supports

the exclusion of the averaged LDEO data with a high standard deviation since it could cause the network to interpret the sub-monthly variability as monthly variability – note that the purpose of this study is to capture the monthly variability.

Like for modeling $A_T$ (Takahashi et al., 2014; Broullón et al., 2019), the Arctic Ocean is one of the regions with the highest RMSE of neural network estimated $TCO_2$. The major Arctic rivers contribute with $TCO_2$ concentrations ranging between 400 and 3600 µmol kg$^{-1}$ (estimated by Tank et al., 2012), derived mainly from carbonated rocks in the watersheds. Other areas like the Okhotsk Sea also show a high RMSE (Table 2 and Fig. 3), probably because of the high riverine input of $TCO_2$ (Watanabe et al., 2009). An input variable accounting for the contribution of the rivers to the $TCO_2$ pool would improve the neural network performance in areas like these, but is not available.

The errors of the $pCO_2$ computed in LDEO with $TCO_2$ from NNGv2LDEO and $A_T$ from NNGv2 (Broullón et al., 2019) are similar to the errors obtained by Landschützer et al. (2014) for the SOCAT database in some of the areas (10-16 µatm, Table 2). This result shows the potential of computing $pCO_2$ values with neural networks trained for other variables of the seawater $CO_2$ system, at least in some ocean regions. The global error of the $pCO_2$ in the LDEO dataset is clearly higher than that obtained by Landschützer et al. (2014) for the SOCAT dataset (22 vs. 12 µatm, respectively), although the critical areas are mainly the same (Fig. S4): equatorial Pacific upwelling system, Arctic and subarctic waters around the Alaska Peninsula, the Southern Ocean, the Gulf Stream and the North Atlantic Current. At this point, the following should be considered: 1) the $pCO_2$ computed in the present study derives from $A_T$ and $TCO_2$ and not from specific modeling for $pCO_2$, and therefore, it contains errors associated to this computation (~6 µatm; Millero, 1995) and to the neural network estimates of $A_T$ and $TCO_2$; 2) the present study includes the Arctic region where the highest errors occur (Table 2; Beaufort Sea and High Arctic areas); and 3) there is a longer temporal range in the present study (1973-2016). The analysis of Landschützer et al. (2014) in the LDEO dataset for data that differs from SOCAT shows a global error higher than the one obtained in the present study for all LDEO data between 1998 and 2011 (25.9 vs. 21.3 µatm, respectively). The error between 40º S-40º N is similar in the two studies (Landschutzer et al. (2014): 16.5 µatm; NNGv2LDEO: 16.4 µatm). Although it is not the main objective of this work, these two last results show how NNGv2LDEO and NNGv2 (Broullón et al., 2019) have the potential to compute $pCO_2$ values between 40º S-40º N with similar errors as the method with the lower error in the $pCO_2$ modeling to obtain a climatology and with lower errors in high latitudes, for the LDEO dataset; even taking into account the inclusion of the critical area of the Arctic in the computation of the error of the $pCO_2$ from the present study (it is not included in Landschützer et al., 2014) and the higher amount of data from high latitudes in the present study (15479 vs. 3799).

### 3.2 Time-series validation

The good generalization of the network in the test dataset containing data from Gv2 and LDEO by the similar RMSE that the one reached in the training set is also evidenced through independent time-series data (Table 3). Except for KERFIX, where the number of data points is very low and Olsen et al. (2019)

suggested an adjustment to the original data of -39 µmol kg$^{-1}$, TCO$_2$ computed using NNGv2LDEO and CANYON-B at the time-series locations are characterized by low errors and biases (Table 3). NNGv2LDEO computes TCO$_2$ with a lower RMSE and bias than CANYON-B for most of the time-series stations (Table 3). CANYON-B reaches a lower RMSE in HOT ALOHA SURFACE and ESTOC than NNGv2LDEO, but the bias is considerably higher in these time series for CANYON-B.

The seasonal variability is well captured by NNGv2LDEO showing its great potential to design a monthly climatology. In the surface layer, where the seasonal variability is the highest, the computed values are strongly correlated with the measured TCO$_2$ in all the time series (Fig. 5). In addition, the high correlation holds for all depths (Table S1). The location of the time series in different oceanographic regimes allows to complement the good TCO$_2$ computation by NNGv2LDEO already shown in the previous independent sets in almost any region of the ocean.

Assessing the potential of neural networks to obtain values of other variables of the seawater CO$_2$ system in the time series, pCO$_2$ calculated with A$_T$ from NNGv2 (Broullón et al., 2019) and TCO$_2$ from NNGv2LDEO compared quite well with pCO$_2$ as measured or calculated from A$_T$ and TCO$_2$ at the time-series stations (Table 4). Except for BATS, the pCO$_2$ obtained in the present study has a lower error than that reported by Landschützer et al. (2014) (Table 4). In contrast, the bias in the present study is higher, except for ESTOC. Considering the error involved in the calculation of pCO$_2$ from A$_T$ and TCO$_2$ (~6 µatm; Millero, 1995) and the error in the computed A$_T$ and TCO$_2$ with the neural networks (Table 4), our results demonstrate again the ability of NNGv2 and NNGv2LDEO to calculate other variables of the seawater CO$_2$ system with a relatively low error.

Using NNGv2LDEO, it is also possible to reproduce the secular trends in TCO$_2$. Using seasonal detrending to enhance the multi-annual changes, similar trends in the longer time series are found for the measured TCO$_2$ and the neural network computed TCO$_2$ (Table 5). The same holds for pCO$_2$ (Table 5), although at the IRMINGER site the trend obtained from the neural network generated data is significantly lower than that from measured data. The neural networks seem to capture the anthropogenic influence in the seawater CO$_2$ system and thus the ocean acidification process (Fig. 6). Furthermore, using NNGv2LDEO increases the amount of TCO$_2$ data where the various inputs were measured but not TCO$_2$ itself. This allows for evaluation of high frequency changes (Fig. 6) and for calculation of interannual trends with a low error (as temporal sampling biases are reduced).

**3.3 Climatology**

Using NNGv2LDEO we have demonstrated its ability to compute TCO$_2$ values with low errors and, especially, to capture the monthly variability of this variable. In addition, the climatologies of the input variables used to create the climatology of TCO$_2$ have been satisfactorily evaluated previously for the construction of an A$_T$ climatology (Broullón et al., 2019). Considering these results, a monthly climatology of TCO$_2$ is obtained by passing the input climatologies through NNGv2LDEO.

The spatial distribution of the surface annual mean climatology of $TCO_2$ (Fig. 7a) is similar to two recent climatologies: those of Takahashi et al. (2014) and Lauvset et al. (2016). The largest surface $TCO_2$ concentrations occur in the Southern Ocean, subpolar North Atlantic, Nordic Seas and Mediterranean Sea (note that the latter is not included in these other climatologies). In general, surface $TCO_2$ decreases from high to low latitudes. The Indian and the Pacific oceans are characterized by lower concentrations of $TCO_2$ at higher latitudes than the Atlantic, the latter being the ocean with the highest surface $TCO_2$ by area. $TCO_2$ increases with depth in all oceans, in particular in the upwelling regions, where this increase is expanded eastwards with depth (Fig. 7b and video in http://dx.doi.org/10.20350/digitalCSIC/10551). Depending on the area, the values reach a maximum at certain intermediate depths and below it the concentration gradually decreases or remains almost constant (Fig. S5).

The largest seasonal variability occurs at the surface in high latitudes, in the Pacific upwelling region, the equatorial African coasts and in the area under influence of the Amazon River (Fig. 8a). At depth, the seasonal variability decreases, except for the Pacific upwelling region where it increases and moves progressively northward between 30 and 150 m (Fig 8b). This increase is correlated with the high seasonal variability of the climatologies of nutrients, oxygen and temperature at these depths. Czeschel et al. (2012) also showed an increase in the subsurface variability of oxygen from measured profiles. Similar increases also occur in the Indian Ocean north of 20º S between 50 and 100 m and in the equatorial Atlantic Ocean in the same depth range. At 1500 m level, the seasonal variability is below 10 µmol kg$^{-1}$ in most of the ocean (Fig. 8c). This last result shows that an annual climatology below 1500 m is sufficient.

Although the surface patterns of the annual mean of the $TCO_2$ climatology are very similar to those of the other recent climatologies (Takahashi et al., 2014; Lauvset et al., 2016), differences do occur. The annual mean climatology of the present study is closest to that of Takahashi et al. (2014) (Table 6). The largest differences between these two climatologies are located in the Arctic, North Pacific, Peru upwelling area, western South Pacific and the area of influence of the Antarctic Circumpolar Current (Fig. S6a). The Atlantic and the Indian oceans do not show significant differences. Our climatology shows more deviations to that of Lauvset et al. (2016), compared in the grid of Takahashi et al. (2014) (Table 6). The highest differences are found in the North Pacific, around Antarctica, Nordic Seas, South and North Atlantic and in several less localized areas around the oceans (Fig. S6b). When the climatology of Takahashi et al. (2014) is compared to that of Lauvset et al. (2016), the differences are even higher (Table 6) and the critical areas are the same of those of the previous comparison. Although it is clear that discrepancies between the three climatologies derive from the different methods used, the higher similarity between ours and the one of Takahashi et al. (2014) is probably due to the influence of the same source used to create them, the World Ocean Atlas.

The comparison of our climatology with that of Lauvset et al. (2016) at the 33 depth levels of Lauvset et al. (2016) shows a reduction of the RMSE with depth. Between 0 and 1000 m, the RMSE is reduced from ~32 to 7 µmol kg$^{-1}$ (Table S2) (note the higher RMSE at surface compared to one obtained for the grid of Takahashi et al. (2014) because of the inclusion of areas which are not included in the latter's grid, and the difficulty of modeling $TCO_2$ in some areas, like the Arctic and the Mediterranean Sea). This reduction with

depth is probably due to the reduction of the variability in most of the ocean below the surface. The surface values in Lauvset et al. (2016) are likely characteristic from months in which most of the sampling was carried out. Because of the lower variability of $TCO_2$ at depth, the values are closer to the annual mean and therefore the two compared climatologies are more similar at depth than in surface depth levels. Below 1000 m, the differences between the two climatologies are not significant, with a RMSE around 5 µmol kg$^{-1}$ and a bias around 0.5 µmol kg$^{-1}$ at each depth level (Table S2).

Our monthly climatology shows a high correspondence with that of Takahashi et al. (2014), although the RMSE values show that there are also large differences in certain areas (Table 7). These areas are mainly the same of those in the comparison of the annual mean climatologies, but some other small regions with high differences appear for each month all through the ocean (Fig. S7).

Unfortunately, the uncertainty of the $TCO_2$ climatology cannot be assessed globally and robustly. As Broullón et al. (2019) stated, the unavailability of an uncertainty field associated to the WOA13 objectively analyzed climatologies does not allow to perform a proper global uncertainty assessment. Therefore, the analysis is relegated to the areas where repeated sampling of $TCO_2$ has been carried out monthly over a long period, that is, the HOT ALOHA and BATS time-series stations. The climatology of $TCO_2$ from NNGv2LDEO is consistent with the monthly climatological values at these two places (Fig. 9). In general, the profiles from the $TCO_2$ climatology are within the variability range (shadow area in Fig. 9) of the monthly averaged measured data for each depth level. In the upper 30 m of the water column, the climatology of $TCO_2$ differs from the measured BATS data from May to August. This difference is mainly explained by the surface error of the network showed for this time series in Fig. 5a, where the computed $TCO_2$ decreases from maximum to minimum sooner than the measured $TCO_2$. For HOT ALOHA, the RMSE of the profiles of the $TCO_2$ climatology oscillates between 3.6 and 9.2 µmol kg$^{-1}$ with a mean value of 6.3 µmol kg$^{-1}$ (bias range: -3.8 to 1.2 µmol kg$^{-1}$; mean bias: -1.4 µmol kg$^{-1}$). At BATS, the RMSE is lower than for HOT ALOHA: 1.1 to 8.5 µmol kg$^{-1}$ with a mean value of 4.4 µmol kg$^{-1}$ (bias range: -1.4 to 7.6 µmol kg$^{-1}$; mean bias: -2.5 µmol kg$^{-1}$). Furthermore, the seasonal variability of the $TCO_2$ climatology is quite similar to that of the measured data at BATS and HOT ALOHA (Fig. S8). Although in other time series there are not enough measured data to obtain climatological values, these pseudo climatological values also correlate very well with the $TCO_2$ climatology (data not shown). These results suggest that the climatology is robust in different oceanographic regimes and adequately captures the seasonal cycle of $TCO_2$.

It has been demonstrated in this study that $pCO_2$ and possibly other variables of the seawater $CO_2$ system can be computed from $A_T$ and $TCO_2$ derived from neural networks with a relatively low error in different datasets (LDEO in Sect. 3.1 and time series in Sect. 3.2). The $pCO_2$ climatology (Fig. S9) computed from the $TCO_2$ climatology of the present study and the $A_T$ climatology of Broullón et al. (2019) is very similar to that of Landschützer et al. (2017). The differences between the annual mean climatology of the two studies are below 15 µatm in most of the ocean (RMSE: 8.3 µatm; bias: 2.9 µatm; r$^2$: 0.82). The differences above this threshold are mainly located in the Pacific equatorial upwelling system, the east part of the South Pacific Gyre, Nordic Seas, Labrador Sea, Atlantic section of the Southern Ocean, Bay of Bengal and the

waters surrounding the east margin of Asia (Fig. S10). In most of these areas, both methods have the greatest
errors (Figs. 2 and 4 in Landschützer et al., 2014 and Fig. S4 of the present study).

On a monthly basis, the RMSE between the two $pCO_2$ climatologies is between 13.6 and 15.6 µatm and
the correlation is lower than for the annual mean comparison ($r^2$: 0.55-0.72 vs 0.82). The areas with the
higher differences are the same as in the annual comparison but other small regions appear along the ocean
month by month (Fig. S11). Furthermore, the seasonal variability in the two climatologies matches in a
great extension of the ocean, although there are areas with notable differences (Fig. S12). In general, the
$pCO_2$ climatology is quite similar to that of Landschützer et al. (2017) and this result contributes to show
that both the $TCO_2$ climatology of the present study and the $A_T$ climatology of Broullón et al. (2019) are
mostly robust and suggest that climatologies of other seawater $CO_2$ system variables can be confidently
computed.

**4 Data availability**

The climatologies of $TCO_2$ and $pCO_2$ and NNGv2LDEO designed in this study are available at the data
repository of the Spanish National Research Council (CSIC; http://dx.doi.org/10.20350/digitalCSIC/10551,
Broullón et al., 2020).

**5 Conclusions**

We presented a tool for computing $TCO_2$ in the global ocean. Compared to previous methods, the
uncertainties in such computations have been reduced. Including two updated datasets containing thousands
of measurements of inorganic carbon variables across the ocean in the training of the neural network, we
were able to capture a wide range of variability of $TCO_2$. The low errors obtained in independent subsets
as in time-series stations, are further evidence of the potential of the network in computing $TCO_2$.

Our global monthly climatology created with a neural network is the first that covers the oceans from the
surface to the abyss at such temporal resolution. In addition to the accuracy of the network, the low
uncertainty of the climatology in different regions and its usefulness in creating climatologies of other
seawater $CO_2$ chemistry variables (i.e. $pCO_2$) show its robustness. Therefore, we present the global
climatology of $TCO_2$ to the scientific community to complement the recently designed climatology of $A_T$
by Broullón et al. (2019) for its use in the initialization and evaluation of models or any other analysis
related to the carbon cycle.

**6 Author contributions**

DB, FFP and AV designed the study. The manuscript was written by DB and revised and discussed by all
the authors. The dataset of the climatology and the neural network were created by DB.

**7 Competing interests**

The authors declare that they have no conflict of interest.

## 8 Acknowledgements

The authors want to thank the comments of the three referees to improve the study. The paper is dedicated to Taro Takahashi, who immensely contributed to the ocean's role in the carbon cycle and passed away in December 2019.

## 9 Financial support

This research was supported by Ministerio de Educación, Cultura y Deporte (FPU grant FPU15/06026), Ministerio de Economía y Competitividad through the ARIOS (CTM2016-76146-C3-1-R) project co-funded by the Fondo Europeo de Desarrollo Regional 2014-2020 (FEDER) and EU Horizon 2020 through the AtlantOS project (grant agreement 633211). Are Olsen was supported by the Norwegian Research Council through ICOS (245927). Mario Hoppema was partly supported by European Union's Horizon 2020 program under grant agreement no. 821001 (SO-CHIC). We acknowledge support of the publication fee by the CSIC Open Access Publication Support Initiative through its Unit of Information Resources for Research (URICI).

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

**Table 1. Differences between the methods used in the present study and in CANYON-B (Bittig et al., 2018).**

| | Bittig et al. (2018) | This study |
|---|---|---|
| Training technique | Bayesian regularization | Levenberg-Marquardt |
| Input variables | Temperature, salinity, oxygen, position and time | Temperature, salinity, oxygen, phosphate, nitrate, silicate, position and time |
| Datasets | GLODAPv2 (Olsen et al., 2016) | GLODAPv2.2019 (Olsen et al., 2019) LDEOv2016 (Takahashi et al., 2017) |

**Table 2. RMSE (bias) by area and depth for $TCO_2$ and $pCO_2$ computed with CANYON-B and NNGv2LDEO in Gv2QC and LDEO datasets. For each depth range, the RMSE (bias) in each area was weighted by the contribution of its data to the total. Units are micromoles per kilogram ($\mu$mol kg$^{-1}$) for $TCO_2$ and microatmospheres ($\mu$atm) for $pCO_2$.**

| | $TCO_2$ | | | | | | | | | | | | $pCO_2$ | |
|---|---|---|---|---|---|---|---|---|---|---|---|---|---|---|
| | 0-50 m | | | | 50-200 m | | 200-500 m | | 500-1000 m | | >1000 m | | 0 m | |
| | LDEO (0 m) | | Gv2QC | | Gv2QC | | Gv2QC | | Gv2QC | | Gv2QC | | LDEO | |
| Areas defined in Takahashi et al. (2014) | NNGv2LDEO | CANYON-B | NNGv2LDEO | CANYON-B | NNGv2LDEO | CANYON-B | NNGv2LDEO | CANYON-B | NNGv2LDEO | CANYON-B | NNGv2LDEO | CANYON-B | NNGv2LDEO | CANYON-B |
| West GIN Seas | 16.8 (7.6) | 21.2 (18.1) | 14.2 (-1.7) | 15.3 (0.7) | 6.4 (-0.6) | 6.7 (0.3) | 5.5 (-1.1) | 6.8 (-0.5) | 4.1 (0.9) | 5 (1.3) | 4 (0.8) | 4.1 (0.8) | 28.6 (17.2) | 34.9 (35.2) |

| Region | | | | | | | | | | | | | | |
|---|---|---|---|---|---|---|---|---|---|---|---|---|---|---|
| East GIN Seas | 11.1 | 9.2 | 10.2 | 11.4 | 5.9 | 6.2 | 4.6 | 5.2 | 3.5 | 4 | 3.9 | 3.7 | 17.3 | 15 |
| | (14.1) | (5.6) | (1.6) | (1.3) | (0.5) | (2.9) | (-0.8) | (1.7) | (-0.6) | (2.3) | (-0.6) | (0.1) | (22.4) | (9.4) |
| High Arctic | 13.3 | 32.4 | 20.2 | 24.1 | 11.4 | 12.3 | 6.5 | 6.6 | 6.8 | 7.6 | 6.1 | 6.2 | 43 | 79.1 |
| | (-8.3) | (13.7) | (-1.1) | (4.4) | (-0.1) | (1.7) | (0.1) | (-0.2) | (-0.1) | (0.6) | (-0.4) | (-1.1) | (-22.8) | (22.6) |
| Beaufort Sea | 29.7 | 42 | 54 | 53.1 | 14.5 | 13.3 | 8.1 | 9.2 | 7.5 | 8.4 | 6.5 | 7.2 | 58.7 | 135.6 |
| | (0.8) | (10) | (0.1) | (1.3) | (-1.7) | (1.8) | (0.1) | (-1) | (2) | (-0.5) | (1.1) | (-2.1) | (6.7) | (5.3) |
| Labrador Sea | 5.3 | 7.5 | 10.5 | 11.6 | 5.4 | 6.6 | 4.1 | 4.3 | 3.5 | 3.7 | 2.7 | 3.1 | 10.7 | 13.6 |
| | (0.6) | (2.7) | (-0.9) | (1.3) | (-0.2) | (1.6) | (-0.6) | (1.4) | (-0.1) | (1.5) | (0.2) | (1.6) | (1) | (4.5) |
| Subarctic Atlantic | 11.7 | 14.6 | 9 | 11.1 | 4.5 | 5.4 | 4.3 | 4.8 | 4.2 | 4.6 | 4 | 4.3 | 21.9 | 26.7 |
| | (3.3) | (11.5) | (-2.3) | (1.5) | (0.1) | (1.9) | (0.1) | (0.3) | (0.1) | (0.4) | (-0.2) | (0) | (7.8) | (23.2) |
| North Atlantic Drift | 11.1 | 12.9 | 13.3 | 14.5 | 9.9 | 10.4 | 7.8 | 7.8 | 4.3 | 4.5 | 3.5 | 3.6 | 21 | 24.7 |
| | (0.5) | (5.7) | (-1.3) | (0.5) | (0.7) | (0.5) | (0) | (-0.4) | (0.2) | (0.1) | (0.3) | (-0.1) | (1.4) | (10.3) |
| Central Atlantic | 7.9 | 9.6 | 15.8 | 14.9 | 6.6 | 6.5 | 5.2 | 5.1 | 4.6 | 4.5 | 4.3 | 4.4 | 13.5 | 17.4 |
| | (-0.3) | (-1.2) | (0) | (0.3) | (0.2) | (1) | (-0.5) | (0.5) | (-0.1) | (0) | (0) | (0) | (-0.3) | (-2) |
| South Atlantic Transition Zone | 7.7 | 13.8 | 7.2 | 7.8 | 5.4 | 5.7 | 5.7 | 4.7 | 5 | 4.8 | 4.3 | 4.2 | 14.6 | 25 |
| | (-1.3) | (-2.1) | (-0.9) | (0.3) | (0.8) | (0.1) | (1.1) | (0.7) | (-0.1) | (1) | (0.5) | (0.3) | (-2.7) | (-5.2) |
| Antarctic Atlantic | 11.8 | 19.2 | 8.6 | 10 | 4.6 | 5.4 | 3.5 | 3.8 | 3.1 | 3.1 | 3.1 | 3.1 | 25.6 | 41.5 |
| | (1.4) | (20.9) | (-1.6) | (0.5) | (0) | (0.4) | (-0.1) | (0.3) | (0.1) | (0.3) | (0) | (0.3) | (4.9) | (50.9) |
| Kuroshio Alaska Gyre | 10.9 | 12.3 | 8.5 | 12.2 | 6.4 | 8.1 | 5 | 5.2 | 4.5 | 4.3 | 3.7 | 3.9 | 20.7 | 23.7 |
| | (1.6) | (1.5) | (-0.9) | (2.4) | (0.9) | (0.3) | (0.3) | (1.1) | (0.6) | (0.3) | (0.4) | (0.3) | (3.7) | (3.4) |
| North Central Pacific | 26.3 | 34.5 | 9.6 | 15 | 6.8 | 8.3 | 4.2 | 4.7 | 4 | 4.1 | 3.4 | 3.8 | 46.7 | 56.6 |
| | (-3.6) | (-9.6) | (0.2) | (3.4) | (0.5) | (0.4) | (0.3) | (0.5) | (-0.3) | (0.3) | (0.3) | (0) | (-0.4) | (-7) |
| Okhotsk Sea | - | - | 23.1 | 16.4 | 11.3 | 6.8 | 6.3 | 5.1 | 5.2 | 3.4 | 4.1 | 3.5 | - | - |
| | - | - | (0.9) | (1.6) | (-1.2) | (-0.7) | (-2.3) | (-1.6) | (-4) | (-1.3) | (1.2) | (1.9) | - | - |
| Central Tropical North Pacific | 8 | 9.7 | 7.9 | 8.8 | 7.2 | 7.2 | 4.9 | 5 | 4.3 | 4.5 | 3.6 | 3.8 | 14.2 | 17.2 |
| | (-1.3) | (-3.2) | (-0.9) | (0.5) | (0.5) | (1.2) | (-0.6) | (0.2) | (-0.4) | (0.5) | (-0.2) | (0.2) | (-2.1) | (-5.6) |
| Tropical East North Pacific | 11.1 | 14.5 | 10.9 | 13.8 | 5.9 | 8.5 | 2.6 | 3.4 | 2.1 | 2.1 | 2.4 | 2.1 | 20.8 | 28.2 |
| | (-0.1) | (-4.5) | (0.9) | (-1.4) | (0.3) | (2.5) | (0.3) | (1.7) | (0) | (0.6) | (-0.2) | (-0.3) | (0.2) | (-8.8) |
| Panama Basin | 12.5 | 17.4 | 10.2 | 9.5 | 6.5 | 3.9 | 4 | 5.8 | 3.8 | 3.2 | 4.2 | 4.3 | 25.8 | 38.8 |
| | (-0.7) | (0.7) | (-3.4) | (1.5) | (-2.7) | (-6.7) | (2.3) | (-1.2) | (0.8) | (1.4) | (0) | (2.8) | (-0.3) | (1.3) |
| Central South Pacific | 10.1 | 12.9 | 10.3 | 10.9 | 8.9 | 9.4 | 4.4 | 4.5 | 3.8 | 3.8 | 3.3 | 3.5 | 18.6 | 24.3 |
| | (-2.1) | (-3.4) | (1.2) | (-0.7) | (0) | (0.2) | (-0.1) | (0.8) | (-0.1) | (-0.4) | (0) | (-0.1) | (-3) | (-5) |
| East Central South Pacific | 10.7 | 15.4 | 10.6 | 15.2 | 6.9 | 7.5 | 4.1 | 2.8 | 3.8 | 3.5 | 3.3 | 3 | 24.1 | 34.2 |
| | (-1) | (-0.1) | (0.6) | (1.4) | (1.2) | (1.3) | (0.4) | (0.1) | (-0.5) | (-0.2) | (-0.6) | (0.3) | (-1.2) | (0.5) |
| Subpolar South Pacific | 6.9 | 7.9 | 5.8 | 7.7 | 5 | 5.4 | 2.9 | 2.8 | 4.7 | 4.8 | 4.1 | 4.4 | 13.7 | 16 |
| | (1.2) | (-0.9) | (-0.7) | (2.4) | (0.4) | (0.6) | (0.9) | (1.7) | (-1) | (1.2) | (0.7) | (1) | (2.4) | (-2.2) |
| Antarctic Pacific | 19.5 | 29.3 | 8.3 | 7 | 3.6 | 4.1 | 2.7 | 3.3 | 2.6 | 2.9 | 2.6 | 2.1 | 34.1 | 53.8 |
| | (1.9) | (4.9) | (-1.4) | (0.5) | (-0.5) | (0.5) | (-0.1) | (0.7) | (0.3) | (0.3) | (0.5) | (-0.1) | (7.7) | (14.9) |
| Main North Indian | 10.8 | 13 | 10.5 | 12.9 | 8.1 | 7.8 | 3.2 | 3.3 | 2.4 | 2.6 | 3.1 | 3.7 | 19.8 | 23.5 |
| | (-1.8) | (-7.7) | (2.8) | (1.7) | (-0.1) | (0.5) | (0.7) | (0.9) | (-0.4) | (-0.4) | (-0.2) | (0.2) | (-2.8) | (-12.7) |
| Red Sea | 18.3 | 20.9 | 12 | 16.8 | 9.4 | 8.7 | 7.6 | 7.9 | 7.4 | 5.7 | 3.3 | 7.2 | 28 | 32.3 |
| | (-13.9) | (-16.7) | (-4.3) | (-3.5) | (0.2) | (-3.7) | (-4.3) | (-5.1) | (1) | (-4.4) | (-1.3) | (-1.1) | (-21) | (-25.5) |
| Bengal Basin | 6 | 3.7 | 9.8 | 7.4 | 7.4 | 6.4 | 2 | 2.1 | 1.9 | 2.1 | 2 | 2.2 | 10.7 | 6.7 |
| | (1.1) | (-5.5) | (-0.2) | (2) | (0.3) | (1.3) | (0.4) | (1.1) | (0.4) | (-0.2) | (-0.6) | (-0.4) | (1.4) | (-10.2) |
| Main South Indian | 8.1 | 10.4 | 9.1 | 10 | 7.1 | 6.9 | 3.8 | 3.8 | 4.2 | 4.5 | 3.4 | 3.8 | 14 | 17.7 |

| | | | | | | | | | | | | | |
|---|---|---|---|---|---|---|---|---|---|---|---|---|---|
| | (-0.2) | (0.4) | (-0.1) | (1) | (0.3) | (0.6) | (-0.1) | (0.9) | (0.1) | (1.1) | (0.1) | (0.1) | (-0.1) | (0.9) |
| South Indian Transition | 8 | 9.4 | 5.3 | 5.3 | 4.1 | 4.4 | 4 | 3.7 | 3.9 | 3.4 | 3.4 | 3.7 | 16 | 18.8 |
| | (-0.4) | (0.7) | (-1.7) | (-0.4) | (0.8) | (0.8) | (0) | (0.5) | (-0.2) | (0.3) | (-0.6) | (-0.9) | (-0.7) | (1.9) |
| Antarctic Indian | 9.7 | 11.8 | 6 | 6.8 | 4.1 | 4.7 | 3 | 3.3 | 2.5 | 2.6 | 2.7 | 2.6 | 23.2 | 28.7 |
| | (1.1) | (5.9) | (-1.4) | (0.2) | (0.4) | (0.6) | (0.1) | (0.3) | (-0.3) | (0.3) | (0.3) | (0.5) | (3.9) | (15.3) |
| Cicumpolar Southern Ocean | 15.9 | 24.5 | 7.6 | 8.2 | 4.4 | 4.9 | 3.1 | 3.4 | 2.9 | 2.9 | 2.9 | 2.7 | 30.7 | 48.9 |
| | (0.9) | (9.8) | (-1.6) | (0.4) | (-0.2) | (0.3) | (-0.1) | (0.2) | (0) | (0.2) | (0.1) | (0.2) | (4.7) | (25.7) |
| Weighted | 11.1 | 15 | 11 | 12.1 | 6.6 | 7 | 4.3 | 4.5 | 4 | 4.1 | 3.5 | 3.7 | 21.1 | 29.6 |
| | (0) | (2.8) | (-0.5) | (0.8) | (0.2) | (0.8) | (-0.1) | (0.5) | (-0.1) | (0.3) | (0.1) | (0.2) | (1.3) | (7.5) |


**Table 3. RMSE and bias between measured and computed TCO$_2$ concentrations in several time series. The comparison was done using only water samples where all the input variables for NNGv2LDEO and the TCO$_2$ were measured in the same water sample.**

| | | | | NNGv2LDEO | | CANYON-B | |
|---|---|---|---|---|---|---|---|
| Time series | Location | Time period | n | RMSE (µmol kg$^{-1}$) | Bias (µmol kg$^{-1}$) | RMSE (µmol kg$^{-1}$) | Bias (µmol kg$^{-1}$) |
| BATS | 31.7ºN, 64.2ºW | 1988-2014 | 4121 | 7.7 | 0.1 | 7.7 | -0.6 |
| HOT ALOHA | 22.8ºN, 158ºW | 1988-2017 | 4054 | 5.4 | -0.5 | 5.1 | -2 |
| HOT ALOHA SURF | 22.8ºN, 158ºW | 1988-2016 | 281 | 6.3 | -1.6 | 5.8 | -5.1 |
| ESTOC | 29.3ºN, 15.5ºW | 1995-2008 | 1697 | 7.1 | 0.8 | 6.6 | 4.7 |
| ICELAND | 68ºN, 12.7ºW | 1985-2013 | 1322 | 5.4 | 5.6 | 6.9 | 5.3 |
| IRMINGER | 64.3ºN, 28ºW | 1991-2013 | 1086 | 4.8 | 3.3 | 7.5 | 6.6 |
| K2 | 47ºN, 160ºE | 1999-2008 | 615 | 3.6 | 1.3 | 6.3 | 2.4 |
| KNOT | 44ºN, 155ºE | 1997-2008 | 1321 | 5.8 | -0.8 | 7.2 | -1.9 |
| OWS | 66ºN, 2ºE | 2001-2007 | 803 | 6.8 | -1 | 10.5 | -4.7 |
| KERFIX | 50.4ºS, 68.2ºE | 1992-1994 | 38 | 13.2 | 26.4 | 13.1 | 28.9 |

**Table 4. RMSE and bias between measured pCO$_2$ (and in some cases, computed from measured A$_T$ and TCO$_2$ in time series where pCO$_2$ was not measured) and computed pCO$_2$ with A$_T$ from NNGv2 (Broullón et al., 2019) and TCO$_2$ from NNGv2LDEO in several time series. The time period for pCO$_2$ from this study is the same as in Table 3. Consult Table 2 in Landschützer et al. (2014) for its time period. The depth range is 0-15 m. Only time series with more than 30 data points are included. RMSE and bias for computed A$_T$ with NNGv2**
**(Broullón et al., 2019) and TCO$_2$ with NNGv2LDEO are included to show the errors in the variables used to compute TCO$_2$.**

| | pCO$_2$ | | | | A$_T$ | | TCO$_2$ | |
|---|---|---|---|---|---|---|---|---|
| | NNGv2LDEO | | Landschützer et al., 2014 | | NNGv2 (Broullón et al. 2019) | | NNGv2LDEO | |
| Time series | RMSE (µatm) | Bias (µatm) | RMSE (µatm) | Bias (µatm) | RMSE (µatm) | Bias (µatm) | RMSE (µatm) | Bias (µatm) |
| BATS | 17.2 | 9.7 | 15.6 | 0.4 | 5.6 | -1.7 | 10.1 | 4.4 |
| HOT ALOHA SURF | 10.3 | -3.6 | 11.6 | 0.1 | 5.0 | 0.9 | 6.5 | -1.6 |
| ESTOC | 10.6 | 2.7 | 14.5 | -7.1 | 2.6 | -2.7 | 5.3 | -0.6 |
| ICELAND | 16 | 14.8 | - | - | 5.4 | 0.7 | 5.4 | 5.4 |
| IRMINGER | 13.1 | -1.8 | 22.6 | -1.1 | 7.0 | -0.4 | 6.6 | -1.1 |

| | | | | | | | |
|---|---|---|---|---|---|---|---|
| K2 | 18.1 | -3.2 | 27.8 | -0.2 | 5.1 | -0.5 | 5.7 | -2.4 |
| KNOT | 20.8 | 8.6 | - | - | 6.6 | -7.3 | 8.2 | -2.5 |

**Table 5. Long-term trends (seasonally detrended) of the measured and computed TCO$_2$ and pCO$_2$ from neural networks at time-series locations in the depth range 0-15 m.**

| | TCO$_2$ (µmol kg$^{-1}$ year$^{-1}$) | | pCO$_2$ (µatm year$^{-1}$) | |
|---|---|---|---|---|
| Time series | Measured | Computed | Measured* | Computed |
| BATS | 1.2 | 1.1 | 1.8 | 1.7 |
| HOT ALOHA SURF | 1.7 | 1.3 | 1.8 | 1.4 |
| ICELAND | 0.9 | 0.9 | 1.5 | 1.6 |
| IRMINGER | 0.6 | 0.5 | 2.5 | 1.7 |

*Computed from measured A$_T$ and TCO$_2$ in time series where pCO$_2$ was not measured.

**Table 6: Comparison of four annual mean surface climatologies of TCO$_2$. Numbers in the lower-left corner represent RMSE. Numbers in the upper-right corner represent r$^2$.**

| RMSE (µmol kg$^{-1}$) - r$^2$ | NNGv2LDEO | Lauvset et al. 2016* | Takahashi et al. 2014 |
|---|---|---|---|
| NNGv2LDEO | - | 0.93 | 0.97 |
| Lauvset et al. 2016* | 19.8 | - | 0.90 |
| Takahashi et al. 2014 | 13.2 | 23.7 | - |

*The domain analyzed is the same as in Takahashi et al. (2014) for coherency reasons.

**Table 7. Comparison of the monthly TCO$_2$ climatology of Takahashi et al. (2014) and the one of the present study.**

| Month | RMSE (µmol kg$^{-1}$) | Bias (µmol kg$^{-1}$) | r$^2$ |
|---|---|---|---|
| January | 16 | 3.0 | 0.95 |
| February | 16.7 | 1.5 | 0.94 |
| March | 15.8 | 2.5 | 0.95 |
| April | 17 | 2.6 | 0.95 |
| May | 16.8 | 2.5 | 0.95 |
| June | 17.2 | 3.2 | 0.95 |
| July | 22.6 | 4.0 | 0.92 |
| August | 17.8 | 3.4 | 0.95 |
| September | 15.5 | 2.5 | 0.97 |
| October | 15.6 | 2.3 | 0.96 |
| November | 15.7 | 2.7 | 0.96 |
| December | 17.6 | 4.3 | 0.95 |

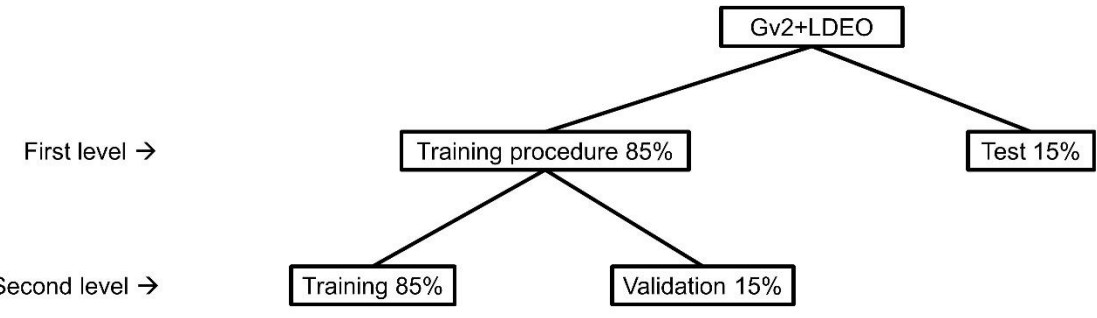

**Figure 1. Division of the complete database in the datasets needed to train the neural network. The percentages in each level are relative to the number of data in the previous one. Data in the datasets of the first level are always the same for each network. Data in the sets of the second level are randomly associated to each set for each network to find the best network weights, because of the different starting points in the error-weight space of the training process (see also Broullón et al., 2019).**

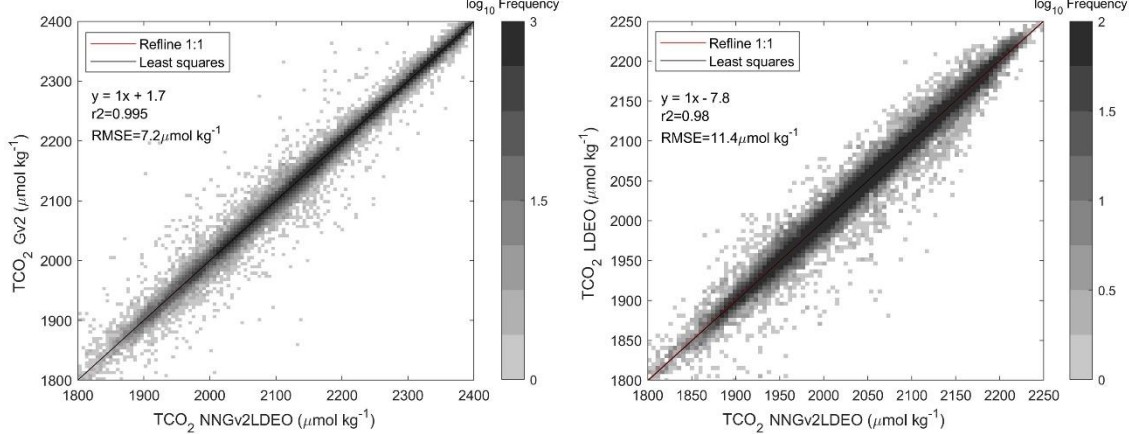

**Figure 2. Regression of TCO₂ computed using NNGv2LDEO and TCO₂ in Gv2 and LDEO. The graph is divided in pixels. The color of each pixel is determined by the number of points inside it. Note the logarithmic scale of the pixels accounting for the large amount of data.**

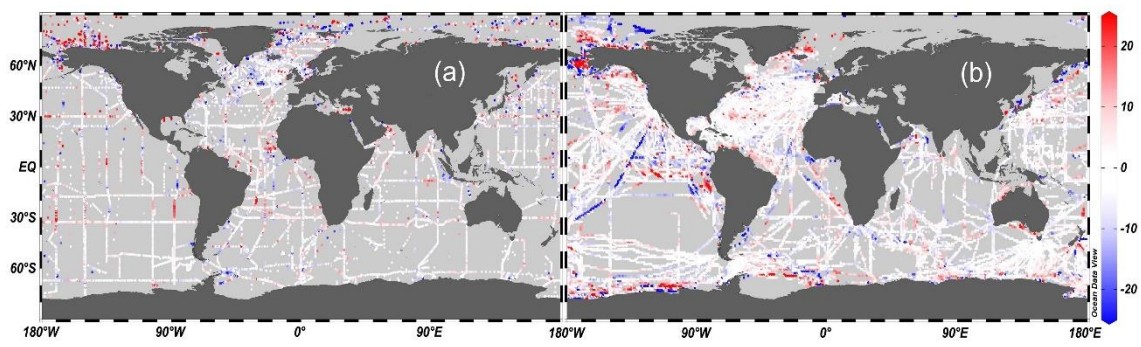

**Figure 3. Differences between (a) Gv2 TCO₂ and NNGv2LDEO TCO₂ (0-30 m) and (b) LDEO TCO₂ and NNGv2LDEO TCO₂ (0 m). This figure was made with Ocean Data View (Schlitzer, 2016).**

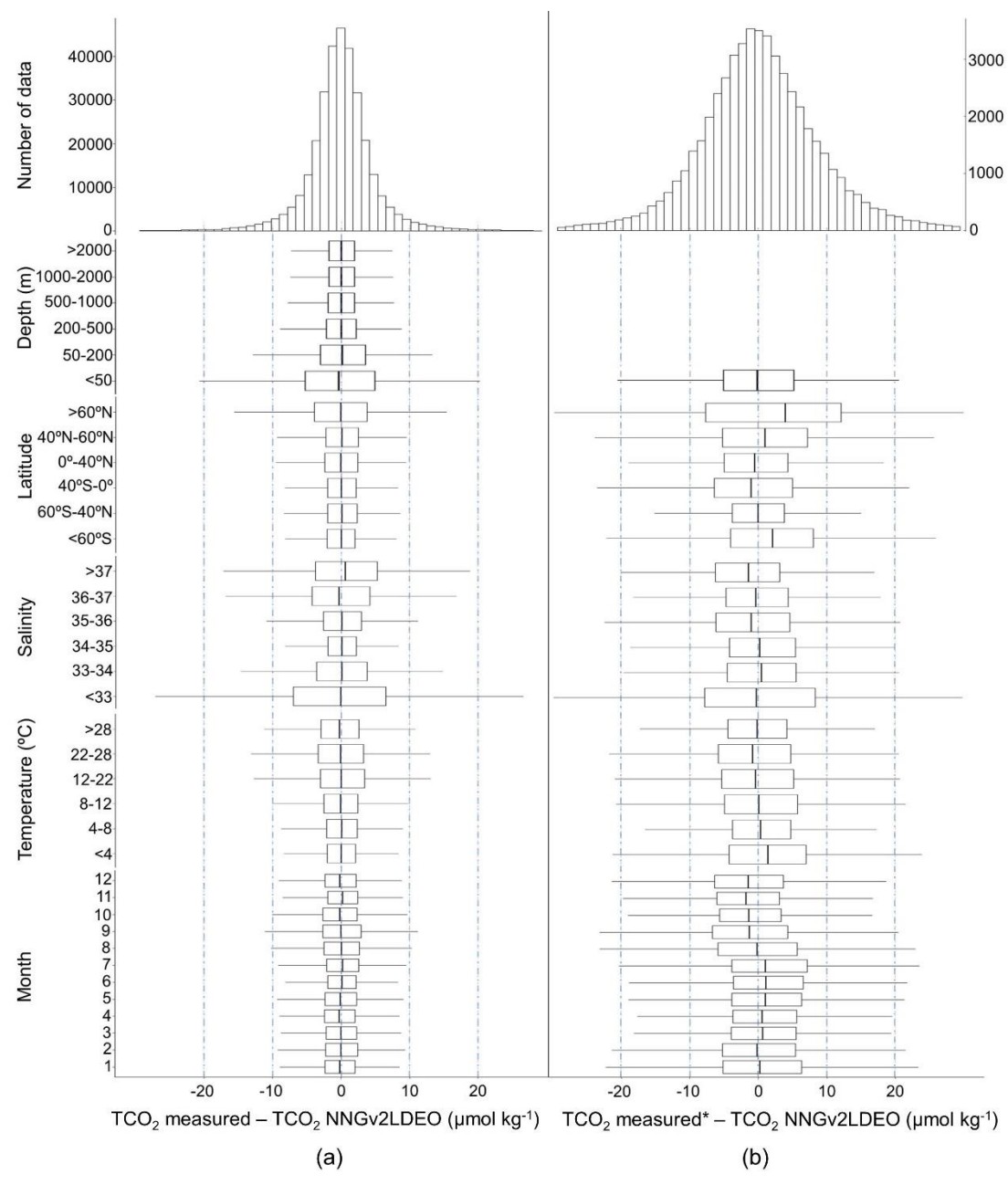

**Figure 4. Histograms and box plots of differences between measured and neural network computed TCO$_2$ in (a) Gv2 and (b) LDEO. *TCO$_2$ computed from measured pCO$_2$ and neural network derived A$_T$.**

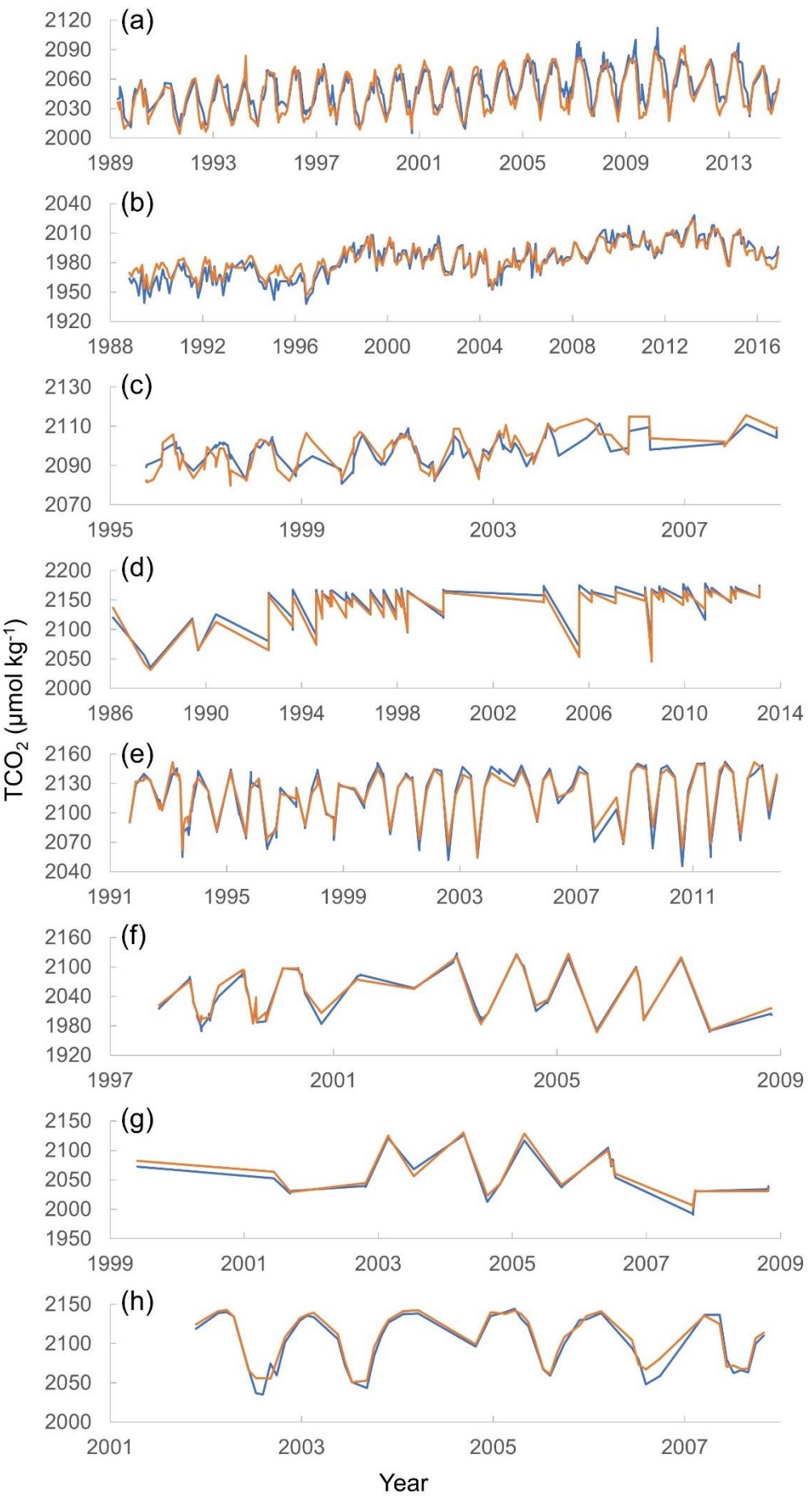

**Figure 5. Measured (blue line) and computed (orange line) TCO₂ with NNGv2LDEO for the depth range 0-15 m (0-30 m in (b)) for several time series. (a) BATS, (b) HOT ALOHA SURFACE, (c) ESTOC, (d) ICELAND, (e) IRMINGER, (f) KNOT, (g) K2 and (h) OWS.**

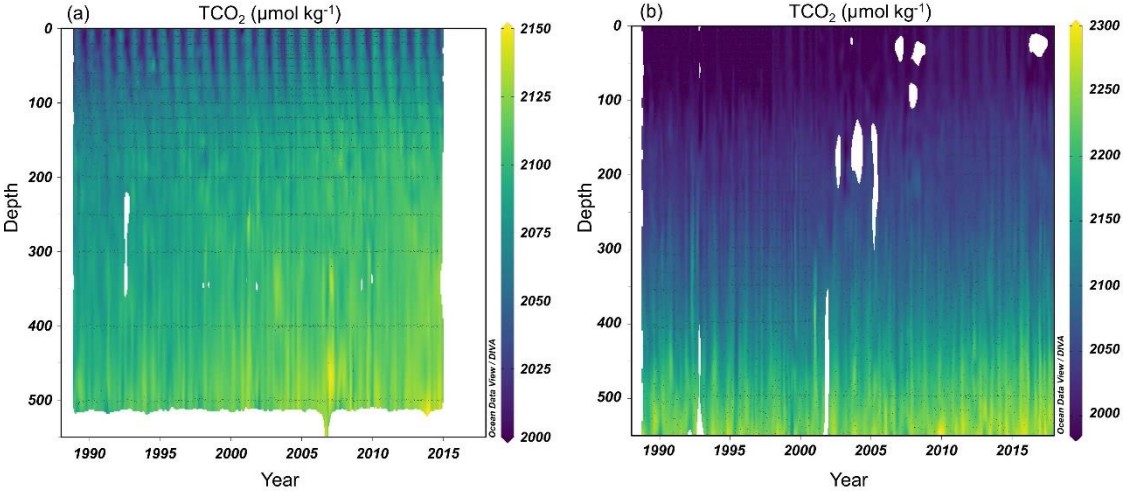

**Figure 6. Time series of TCO₂ using NNGv2LDEO at (a) BATS and (b) HOT ALOHA locations. The water column shows a higher concentration of TCO₂ year by year. This figure was made with Ocean Data View (Schlitzer, 2016).**

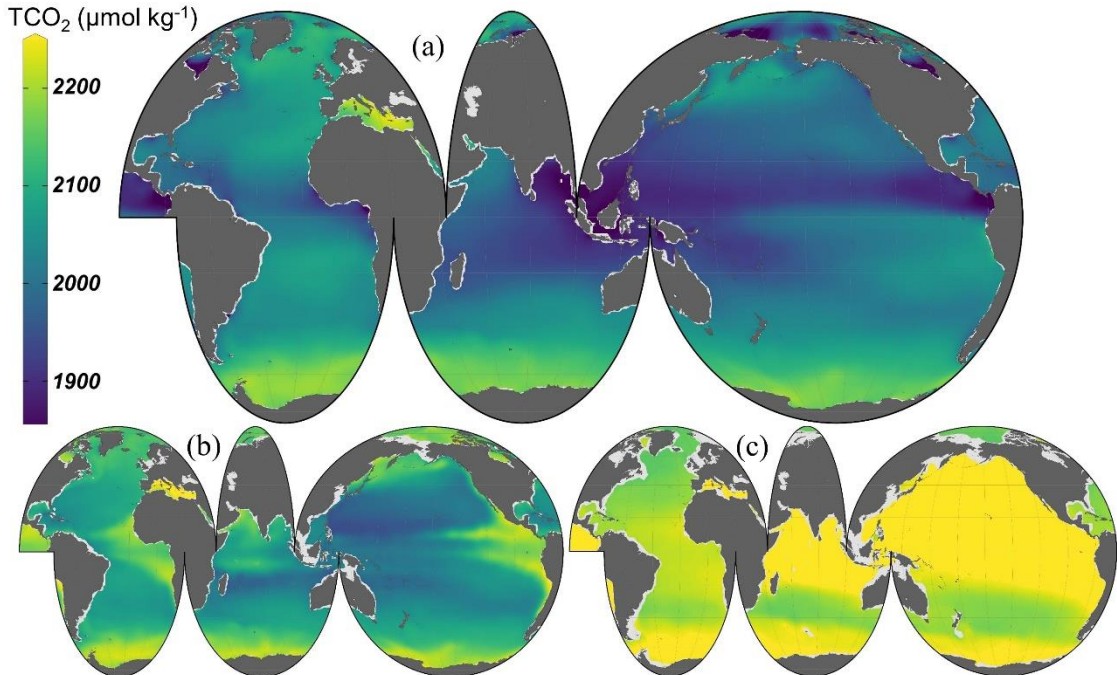

**Figure 7. Annual mean climatology of TCO₂ at (a) 0 m, (b) 100 m and (c) 1000 m. This figure was made with Ocean Data View (Schlitzer, 2016).**

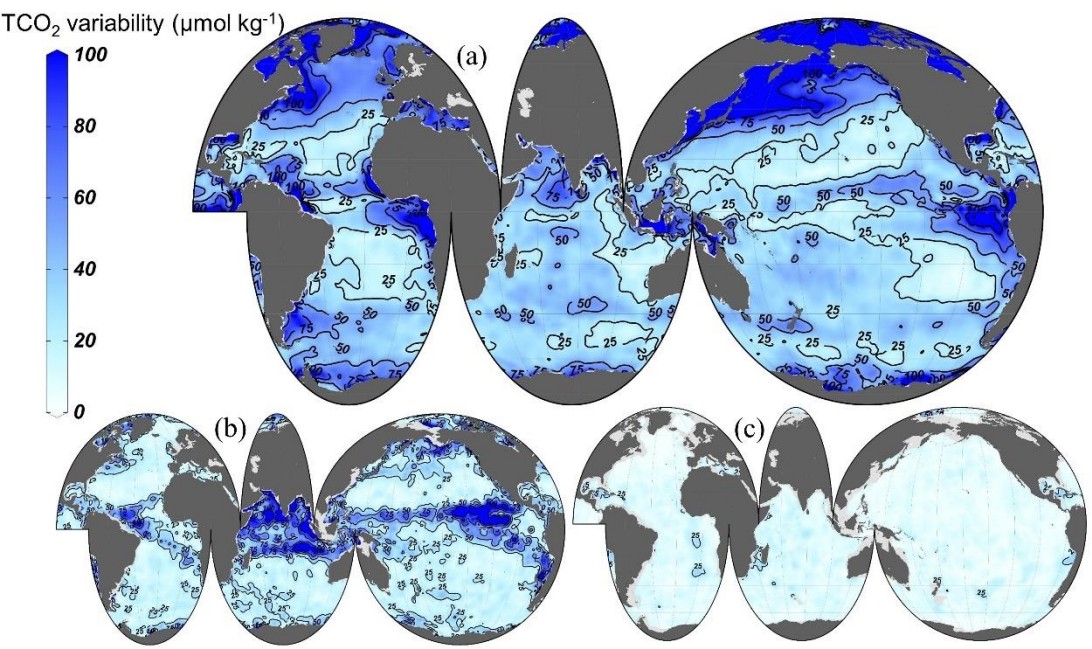

Figure 8. Seasonal amplitude of TCO$_2$ at (a) 0 m, (b) 100 m and (c) 1500 m. The contour lines of 25, 50, 75 and 100 µmol kg$^{-1}$ are shown. This figure was made with Ocean Data View (Schlitzer, 2016).

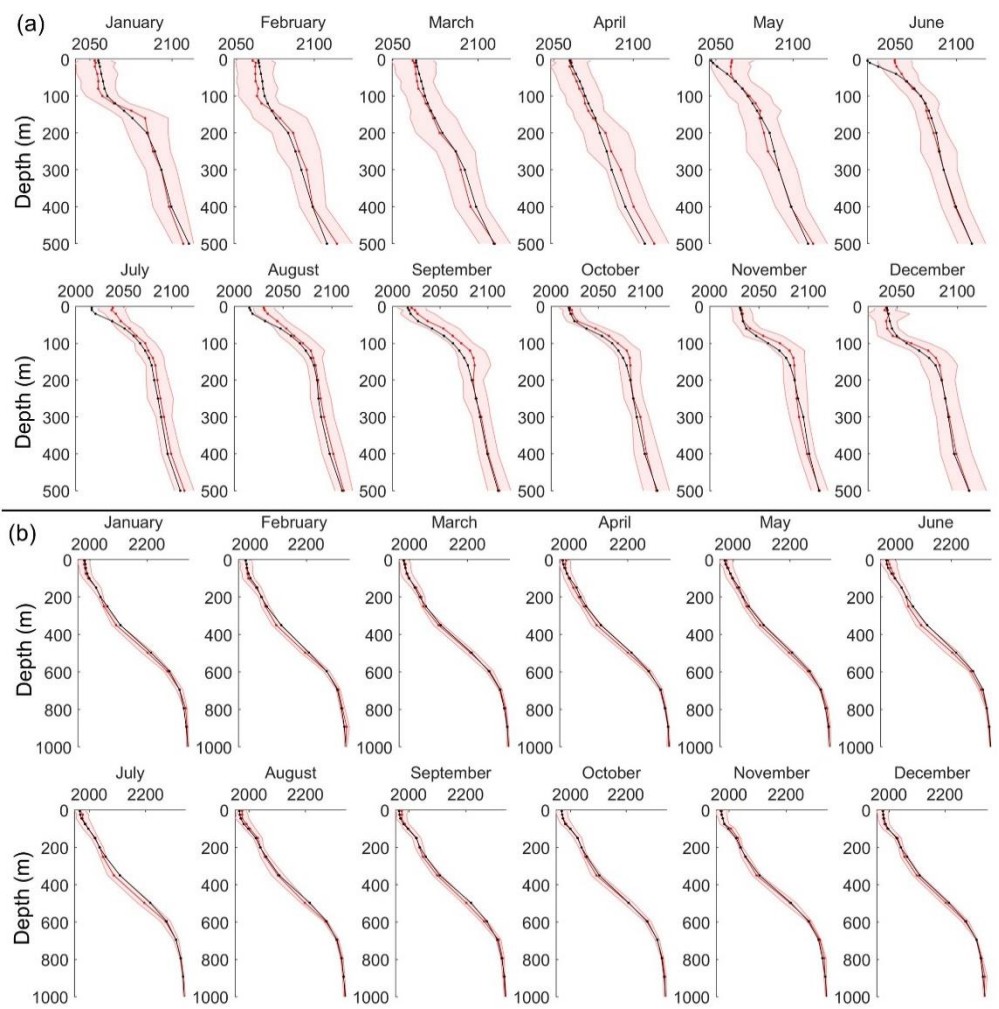

Figure 9. Comparison of the monthly climatological profiles of TCO$_2$ computed from measured data (red profile; shadow area is the standard deviation of the averaged values at each depth level) and those from the

815 **TCO$_2$ climatology at (a) BATS and (b) HOT ALOHA locations. Units on the x axis are micromoles per kilogram ($\mu$mol kg$^{-1}$).**