# Peer review of "A global monthly climatology of oceanic total dissolved inorganic carbon: a neural network approach"

_Earth System Science Data, 2020_

## Referee Comment (RC1) · Anonymous Referee #1 · 21 May 2020

The authors describe a neural network approach to obtain TCO2 from geolocation, depth, temperature, salinity, oxygen together with nutrients (nitrates, phosphates, silicates). This approach is trained on GLODAPv2.2019 and LDEO data, therefore with a good surface and in-depth amount of data. Using this neural network, NNGv2LDEO, they produce a monthly TCO2 climatology for the global ocean at different depths. Additionally, they also produce a surface pCO2 climatology.

The manuscript is globally well-structured and understandable and the associated clmatologies could be useful for modelers. However, a few elements still require clarification.

- In section 2.1, "an additional criterion based on the influence of each input variable on the TCO2 extracted" is mentioned. More details on that criterion and its results would be appreciated.

- In section 3.1, when justifying the choice to continue with NNGv2LDEO over NNGv2QCLDEO, you highlight that Gv2 has more data points than Gv2QC in the Mediterranean Sea for example. While these additional data are important for the quality and spatial coverage of your climatology, the absence of these data in Gv2QC might be because of their poor quality. How do you thus justify their use? Additional data in an underrepresented area is a good thing but if these data are of poor quality, does it really constitute an improvement? It would be interesting to compute the errors of NNGv2LDEO specifically in the areas not in Gv2QC. This is kind of visible in Figure 3 but could gain from being developed.

- In section 3.2, it is mentioned that "the high correlation holds for all depths". It would be interesting to see the variations in these correlations according to depth layers.

- In the dataset provided, there are some extremely high pCO2 values (>5000, North of Russia). These are not addressed in the manuscript ad absolutely should be. When plotting the data without tuning the colorbar, it's the first thing a user sees. If stemming from an error, please consider correcting the dataset. If not, this should be described in the manuscript as to provide the data user with a warning of these extreme/erroneous data.

- What is the monthly distribution of the data used and thus how does this affect the monthly climatology (i.e. underrepresentation of winter months leading to a lower confidence in the data product)? - A pCO2 climatology product is created and compared to other existing products. Several figures (including the supplementary material) highlight these comparisons but it would be useful to have a general climatology representation (something like Fig 7). This would avoid/nuance the problem mentioned earlier of the extreme values. - Be aware that the Figures 6, 7 and S4 are not readable by a

colour-deficient individual. Please consider changing the colormap chosen.

- L65: "being these changes reflected" -> these changes being reflected

- L105: the list of predictor variables is missing the date of sampling

- L132: "minimize de errors", remove de

- L133: wrong spelling of author name

- L255: "The variable selected (...) was that labeled as spco2_raw". Please indicate the complete name of the variable, or what it refers to

- L397: "are the same of as those", remove of

- Table 6: The RMSE vs $r^2$ table is kind of hard to read. Maybe find another way to arrange it. Or describe it more in the title.

- Figure 1: "first level are always the same for each of network", remove of

- Figure 5: NGv2LDEO -> NNGv2LDEO; HOL ALOHA SURFACE -> HOT ALOHA SURFACE

- Figure S5: spelling of the word climatology

- Figure S8-S9: spelling of the word microatmospheres

[Figure]

---

## Referee Comment (RC2) · Anonymous Referee #2 · 22 May 2020

This work applied the method of Broullon et al. (2019) to TCO2 and extended the NN model by including year as an input and including TCO2 computed from LDEO pCO2 in the target. The manuscript is clearly written except for a few elements that require clarification; and the climatology TCO2 data are useful for other modelers.

General Comments:

While including LDEO is expected to improve modeling TCO2 dependence on input variables in the surface waters, it raises two questions. The first is the increase of the spatially biased sampling, which could lead to model optimization more weighted toward fitting the surface measurements. The second is the unknown system bias

of the computed TCO2 relative to GLODAPv2 TCO2. This bias could be estimated if there were enough overlapped points within the space and time resolutions of the training data. If you do the regression of Fig.2a using only the surface data, you may detect the bias. As the regression slopes of Fig.2a and 2b are 1, subtraction of the two predictions indicates the system bias of computed LDEO TCO2. You mentioned on line 280 that "Interestingly, CANYON-B is able to reproduce the TCO2 data derived from the complete LDEO dataset with a lower error than the one it obtains for the complete Gv2QC dataset in the surface ocean...". Another explanation to this is that because of the unbiased nature of a NN model (the overall prediction error is close to zero), the system bias of LDEO TCO2 could happen to fall between the prediction biases of Gv2 TCO2 in the surface and interior.

Absolute errors are often used in tables and figures. They hide the information whether the errors show under-estimate or over-estimate; Therefor showing negative errors are recommended.

Specific Comments:

Line 132: The reference of Rumelhart et al. (1986) is missing.

Line 149-152: Could you give more details on how to ensure biogeochemical variables have a larger influence than position variables?

Line 254: Why average 1981 to 2015 to obtain 1995 climatology? You have 20 year from 1995 to 2015, but only 14 year from 1981 to 2015.

Line 261: This is an important criterion to select the NN for making prediction, but no detail available. Could you supply more information in the supplement material on the influences of position variable of the networks?

Table 2. Are the errors absolute? If so, please state explicitly. Also, the global errors should be added. Showing negative errors are more meaningful.

Table 6: Does the label "NNGv2" means NNGv2LDEO?

Table 7: Is the bias absolute? If so, please state explicitly. The global errors should be added. Showing negative errors are more meaningful.

Figure 1b: "y=1x +- 7.8" should be y=1x - 7.8.

Figure 3: Showing negative errors are more meaningful.

Figure 4b: The error bar for depth < 50m should be added using the surface errors.

Figure S1.b: There should be a "+" operator between bj*a0 and SUM(wij*ai) in the activation function.

Figure S2: How the std is calculated for T, S, and pCO2. Modelled TCO2 is larger than observed TCO2 for all pCO2 STD > 4. How to explain this?

Figure S3. If the difference is absolute, please state clearly. Showing negative errors are more meaningful.

Figure S5 and S6: Showing negative errors are more meaningful.

Figure S7a: The model produces a much larger seasonal amplitude in the surface water. Unless measurements are not available in all months, the seasonal amplitude of the climatology should be no larger than that of the measurements. Does this indicates either over-fitting or extrapolation in seasons of no measurements.

Figure S8 and S9. Plotting land with colors confuses grasping the contours of differences.

---

## Referee Comment (RC3) · Anonymous Referee #3 · 26 May 2020

The authors provide a nice description of global TCO2 distribution based on monthly climatology from a neural network approach. The addition input variable "year" is reasonable and important for TCO2. It is great to see that the neural network outputs match very well with the measured TCO2 for the independent time series locations. This manuscript is well organized and easy to follow. I would like to see the publication of this work.

Below are specific comments.

Lines 71-74, include the influence of mixing on TCO2 variability. For example, upwelling-induced increase in TCO2, which is described in lines 83-86.

[Figure]

Lines 75-76, the temperature and salinity influence on TCO2 through the modification of the solubility of CO2 affects the seawater pCO2 (which is almost instantaneous) and thus the air-sea CO2 flux, which eventually drives the change in TCO2 over time.

Line 131, delete de in "minimize de errors"

Lines 142-143, move the full list of the input variables to line 153 to make it ahead of the sentence, "In addition to the target variable. . ."

Line 147, specify which generalization method was used to prevent overfitting.

Line 258, provide how many hidden layers are used in the neural network

Line 263, the errors cannot be avoided in any case. It is likely that a much smaller weight is assigned to a variable that contains a large error.

Line 276, provide how depth-weighted RMSE is calculated

Lines 349-351, the error may come from the bias in total alkalinity. I agree that the error involved in the calculation of pCO2 from AT and TCO2 can be large, but the monthly averaged error is another case. In addition, the choice of K1 and K2 affects the calculated pCO2. Lueker et al. (2000) is a better choice in the case of calculating pCO2. Provide the bias information for total alkalinity in each region in Table 4.

Line 420-430, Fig. 9, the differences in the surface water at BATS in May, June, July, and August seem large as compared to the values in other months. Provide an explanation.

Line 445-448, the bias for pCO2, considering it is the averaged value over space and time, seems large as compared to the bias in Landschutzer et al. (2017) as shown in Table 4. Therefore, I don't agree that climatologies of other seawater CO2 system variables can be confidently computed. Furthermore, this statement compromises the value of this study if TCO2 can be confidently computed from pCO2 and AT.

Fig. 2, color-coded depth instead of the log10 frequency would provide more information regarding the distribution of errors. Fig. S4, add another panel with a similar generated from measured bottle TCO2 data. Furthermore, add one more panel to show the differences between the derived TCO2 and the bottle TCO2. The differences are partially shown in Table 2 with the average differences by region. An additional panel regarding the differences would help to determine the reliability of this new dataset on a smaller scale (e.g., 1 degree by 1 degree).
* * *

---

## Author Comment (AC1) · 18 Jun 2020

**RC**: Referee comment; **AR**: author's response; **AC**: author's changes in manuscript.

**Referee #1 response**

**RC**: The authors describe a neural network approach to obtain TCO2 from geolocation, depth, temperature, salinity, oxygen together with nutrients (nitrates, phosphates, silicates). This approach is trained on GLODAPv2.2019 and LDEO data, therefore with a good surface and in-depth amount of data. Using this neural network, NNGv2LDEO, they produce a monthly TCO2 climatology for the global ocean at different depths. Additionally, they also produce a surface pCO2 climatology.

The manuscript is globally well-structured and understandable and the associated climatologies could be useful for modelers. However, a few elements still require clarification.

**AR**: Thank you so much for the thorough revision of the manuscript. We are pleased to see that you think the climatology could be useful for modelers. We hope to clarify your comments with the answers we give in this document and improve the manuscript. At the end of the answers, the new version of the manuscript is attached for a global view. We have tried to keep a balance to respond appropriately to the 3 reviewers.

**RC**: In section 2.1, "an additional criterion based on the influence of each input variable on the TCO2 extracted" is mentioned. More details on that criterion and its results would be appreciated.

**AR**: We have added more details at the end of section 2.1 and the results in section 3.1 and Fig. S2.

**AC**: In section 2.1: "The influence of each input variable on the computed $TCO_2$ was obtained from Eq. (1):

$$C_i = \sum_{k=1}^{H} w_{ik} \cdot w_k$$

where $C_i$ is the relative importance of the input variable i, H the number of neurons in the hidden layer, $w_{ik}$ the weight of the connection between the variable i and the neuron k of

the hidden layer and $w_k$ is the weight of the connection between the neuron k of the hidden layer and output layer."

In section 3.1: "For this network, the influence of each input variable on the computed TCO$_2$ is depicted in Fig. S2. The position variables together (latitude, clongitude, slongitude and depth) have no more than 30% influence, allowing biogeochemical variables to be the main ones responsible for the variability of TCO$_2$. Furthermore, the input variable year has an influence lower than 5%. This is probably responsible for capturing the positive interannual trend due to the TCO$_2$ increase derived from anthropogenic emissions of CO$_2$ to the atmosphere (see Sect. 3.2)."

[Figure]

Figure S2. The relative importance of the input variables for NNGv2LDEO. lat: latitude; clon: clongitude; slon: slongitude; temp: temperature; sal: salinity; phosp: phosphate; nit: nitrate; sil: silicate; oxy: oxygen.

**RC**: In section 3.1, when justifying the choice to continue with NNGv2LDEO over NNGv2QCLDEO, you highlight that Gv2 has more data points than Gv2QC in the Mediterranean Sea for example. While these additional data are important for the quality and spatial coverage of your climatology, the absence of these data in Gv2QC might be because of their poor quality. How do you thus justify their use? Additional data in an

underrepresented area is a good thing but if these data are of poor quality, does it really constitute an improvement? It would be interesting to compute the errors of NNGv2LDEO specifically in the areas not in Gv2QC. This is kind of visible in Figure 3 but could gain from being developed.

**AR**: The data from the Mediterranean Sea are not included in the QC dataset because of the impossibility to make the crossover analysis, not because of a poor quality of the data. Therefore, we decided to include this kind of data from Gv2 in our study and let the network avoid the possible errors in data, that is, testing its generalization in independent datasets to show there is no overfitting towards possible bad quality data.

We have included the statistics in the noQC dataset by depth.

**AC**: In section 3.1: "For data from Gv2 where no QC was performed for at least one of the variables used in the present study (Gv2noQC), the RMSE also decreases with increasing depth: <50 m: 22.5 µmol kg$^{-1}$; 50-200 m: 9.8 µmol kg$^{-1}$; 200-500 m: 7 µmol kg$^{-1}$; 500-1000 m: 5.4 µmol kg$^{-1}$; >1000 m: 5.4 µmol kg$^{-1}$. Thus, the error in Gv2noQC is similar to that in the areas with the highest error in Gv2QC (Table 2; except in Beaufort Sea, where the error is considerably higher). However, the higher error in Gv2noQC is mainly caused by the samples located in the Arctic Ocean, since cruises in the Atlantic and Pacific oceans are modeled with a very low error. Therefore, using Gv2noQC does not imply introduction of low-quality data in our study, otherwise the network would not compute TCO$_2$ with low errors in Gv2QC because of an overfitting of the possible low-quality data that Gv2noQC could contain."

**RC**: In section 3.2, it is mentioned that "the high correlation holds for all depths". It would be interesting to see the variations in these correlations according to depth layers.

**AR**: A table with the error by depth has been added to the supplementary document.

**AC**:

Table S1. RMSE (bias) between measured and computed TCO$_2$ concentrations in several time series. Units are micromole per kilogram (µmol kg$^{-1}$). *There is only one sample below 500 m.

| Depth range | BATS | HOT ALOHA | ESTOC | ICELAND | IRMINGER | K2 | KNOT | OWS | KERFIX |
|---|---|---|---|---|---|---|---|---|---|

| | | | | | | | | | |
|---|---|---|---|---|---|---|---|---|---|
| 0-50 m | 8.7 (2.7) | 5.8 (-2.6) | 4.4 (-0.2) | 7.8 (7.1) | 5.3 (-0.1) | 5.3 (0.3) | 8.5 (-1.3) | 10.1 (-5.3) | 10 (23.9) |
| 50-200 m | 7.4 (-0.3) | 6.2 (0.5) | 7.7 (3.9) | 4.8 (7.5) | 4.9 (4.9) | 3.6 (1.4) | 5.4 (-0.9) | 7.3 (-1.2) | 10 (26.4) |
| 200-500 m | 6.1 (-1.8) | 4.2 (0.9) | 5.9 (1.9) | 3.1 (6.2) | 3.2 (5) | 2.6 (1.2) | 6 (-1.1) | 3.9 (-1) | 19.2 (28.8) |
| >500 m | 6.4 (-2.7) | 3.8 (-1.6) | 6.6 (-0.6) | 3.4 (3.3) | 3.5 (5.5) | 2.1 (1.7) | 4.1 (-0.4) | 3.1 (1.9) | -* |

**RC**: In the dataset provided, there are some extremely high pCO2 values (>5000, North of Russia). These are not addressed in the manuscript ad absolutely should be. When plotting the data without tuning the colorbar, it's the first thing a user sees. If stemming from an error, please consider correcting the dataset. If not, this should be described in the manuscript as to provide the data user with a warning of these extreme/erroneous data. A $pCO_2$ climatology product is created and compared to other existing products. Several figures (including the supplementary material) highlight these comparisons but it would be useful to have a general climatology representation (something like Fig 7). This would avoid/nuance the problem mentioned earlier of the extreme values.

**AR**: We have included the figure of the annual mean $pCO_2$ climatology with a warning about the extremely high $pCO_2$ values, which come from a $TCO_2/A_T$ ratio higher than 1. In the coastal waters of the Arctic Ocean in general, the low salinities determine low values of $TCO_2$ and $A_T$ in the measured data and in the computed climatologies. Although the neural networks are able to compute low values of $TCO_2$ and $A_T$, they are not able to model the high concentrations of both variables carried by the Arctic rivers to the coastal waters, given this uncertainty a $TCO_2/A_T$ ratio higher than 1 and the consequent extremely high $pCO_2$ values.

**AC**:

[Figure]

Figure S9. Annual mean $pCO_2$ centered in 1995 computed from the annual mean $A_T$ (Broullón et al., 2019) and $TCO_2$ (this study). It should be noted that the extremely high values in the coastal waters of the Arctic Ocean are derived from a $TCO_2/A_T$ ratio higher than 1. This fact is determined by the difficulty of neural networks to model both variables under the influence of river discharges with high concentrations of $A_T$ and $TCO_2$, but $TCO_2/A_T$ ratios higher than 1 are also found in the Arctic Ocean for the measured data in GLODAPv2.2019 (Olsen et al., 2019).

**RC**: What is the monthly distribution of the data used and thus how does this affect the monthly climatology (i.e. underrepresentation of winter months leading to a lower confidence in the data product)?

**AR**: The lack of winter data in some regions is not moving the climatology to an underrepresentation of winter conditions. Vázquez-Rodriguez et al. (2012) demonstrated for the Atlantic Ocean that the subsurface reference layer preserves the surface winter conditions for $CO_2$ system variables in other seasons. We have demonstrated this fact in Broullón et al. (2019) (Table 5) for all the oceans training a neural network with no winter data and finding that the errors in winter data were of the same order of magnitude as the ones obtained with the neural network trained with all Gv2 data. Therefore, neural networks are able to know the surface winter relationships between the used variables without needing winter data, probably thanks to the data in the subsurface reference layer at other seasons. Furthermore, in the present study, we have included the LDEO database which considerably increase the surface winter data.

**RC**: Be aware that the Figures 6, 7 and S4 are not readable by a colour-deficient individual. Please consider changing the colormap chosen.

**AR**: Thanks. Changed.

**AC**:

[Figure]

Figure 6

[Figure]

Figure 7

[Figure]

Figure S4

RC: L65: "being these changes reflected" -> these changes being reflected

AR: Changed.

RC: L105: the list of predictor variables is missing the date of sampling

AR: Thanks. Added.

RC: L132: "minimize de errors", remove de

AR: Thanks. Removed.

RC: L133: wrong spelling of author name

**AR**: Thanks. Changed.

**RC**: L255: "The variable selected (…) was that labeled as spco2_raw". Please indicate the complete name of the variable, or what it refers to

**AR**: Added clarification.

**AC**: "… was that labeled as spco2_raw (sea surface pCO2) in the netCDF file."

**RC**: L397: "are the same of as those", remove of

**AR**: Thanks. Removed.

**RC**: Table 6: The RMSE vs r2 table is kind of hard to read. Maybe find another way to arrange it. Or describe it more in the title.

**AR**: Changed.

**AC**:

**Table 6: Comparison of four annual mean surface climatologies of TCO$_2$. In the lower-left corner is RMSE. In the upper-right corner is r$^2$.**

| RMSE ($\mu$mol kg$^{-1}$) - r$^2$ | NNGv2LDEO | Lauvset et al. 2016* | Takahashi et al. 2014 |
|---|---|---|---|
| NNGv2LDEO | - | 0.93 | 0.97 |
| Lauvset et al. 2016* | 19.8 | - | 0.90 |
| Takahashi et al. 2014 | 13.2 | 23.7 | - |

*The domain analyzed is the same as in Takahashi et al. (2014) for coherency reasons.

**RC**: Figure 1: "first level are always the same for each of network", remove of

**AR**: Thanks. Removed.

**RC**: Figure 5: NGv2LDEO -> NNGv2LDEO; HOL ALOHA SURFACE -> HOT ALOHA SURFACE

**AR**: Thanks. Corrected.

**RC**: Figure S5: spelling of the word climatology

**AR**: Thanks. Changed.

**RC**: Figure S8-S9: spelling of the word microatmospheres

**AR**: Thanks. Changed

[revised manuscript text omitted]

Figure S1. (a) Neural network configuration. The notation is in agreement with Hagan et al. (2014). **a**: input vectors; **W**: weight matrix; **b**: bias matrix; ∑: sum; f: transfer function; **a**$^X$: output matrix. The superscripts indicate the number of the layer. $cLongitude = cos(\frac{\pi}{180°}longitude)$; $sLongitude = sin(\frac{\pi}{180°}longitude)$. The dimensions of the matrices are for an individual sample. Modified from Hagan et al. (2014). (b) Neuron. $a_i$: inputs to each neuron; $w_{i,j}$: weights of each input to each neuron. Modified from Russell and Norvig et al. (2010).

[Figure]

Figure S2. The relative importance of the input variables for NNGv2LDEO obtained with Eq. (1). lat: latitude; clon: clongitude; slon: slongitude; temp: temperature; sal: salinity; phosp: phosphate; nit: nitrate; sil: silicate; oxy: oxygen.

[Figure]

Figure S3. Box plots of differences between measured and computed TCO$_2$ in LDEO by standard deviation (std) ranges obtained from the monthly average of the LDEO data.

[revised manuscript text omitted]

---

## Author Comment (AC2) · 18 Jun 2020

RC: Referee comment; AR: author's response; AC: author's changes in manuscript.

**Referee #2 response**

RC: This work applied the method of Broullón et al. (2019) to TCO2 and extended the NN model by including year as an input and including TCO2 computed from LDEO pCO2 in the target. The manuscript is clearly written except for a few elements that require clarification; and the climatology TCO2 data are useful for other modelers.

AR: Thank you so much for the thorough revision of the manuscript. We are pleased to see that you find the climatology useful for modelers. We hope to clarify your comments with the answers we give in this document and improve the manuscript. At the end of the answers, the new version of the manuscript is attached for a global view. We have tried to keep a balance to respond appropriately to the 3 reviewers.

RC: While including LDEO is expected to improve modeling TCO2 dependence on input variables in the surface waters, it raises two questions. The first is the increase of the spatially biased sampling, which could lead to model optimization more weighted toward fitting the surface measurements.

AR: The inclusion of the variable depth as a predictor and its relevance in the computed $TCO_2$ (see Fig. S2) overcome this possible problem, otherwise errors below the surface (see Table 2) would be considerably higher than the measurement uncertainty in Gv2 (4 µmol kg$^{-1}$ for $TCO_2$; Olsen et al., 2016).

RC: The second is the unknown system bias of the computed TCO2 relative to GLODAPv2 TCO2. This bias could be estimated if there were enough overlapped points within the space and time resolutions of the training data. If you do the regression of Fig.2a using only the surface data, you may detect the bias. As the regression slopes of Fig.2a and 2b are 1, subtraction of the two predictions indicates the system bias of computed LDEO TCO2. You mentioned on line 280 that "Interestingly, CANYON-B is able to reproduce the TCO2 data derived from the complete LDEO dataset with a lower error than the one it obtains for the complete Gv2QC dataset in the surface ocean…".

Another explanation to this is that because of the unbiased nature of a NN model (the overall prediction error is close to zero), the system bias of LDEO TCO2 could happen to fall between the prediction biases of Gv2 TCO2 in the surface and interior.

**AR**: We have now added the bias by area, depth, dataset and method to Table 2.

**AC**:

[revised manuscript text omitted]

**RC**: Absolute errors are often used in tables and figures. They hide the information whether the errors show under-estimate or over-estimate; Therefore, showing negative errors are recommended.

**AR**: We have added the bias in all the tables where RMSE is computed. The figures where absolute differences were shown have been changed for the equivalent with the differences to show negative errors.

**RC**: Line 132: The reference of Rumelhart et al. (1986) is missing.

**AR**: Thanks. Added.

**RC**: Line 149-152: Could you give more details on how to ensure biogeochemical variables have a larger influence than position variables?

**AR**: We have added more information about the method to extract the influence of each input variable in the computed TCO2.

**AC**: In Section 2.1: "The influence of each input variable on the computed $TCO_2$ was obtained from Eq. (1):

$$C_i = \sum_{k=1}^{H} w_{ik} \cdot w_k$$

where $C_i$ is the relative importance of the input variable i, H the number of neurons in the hidden layer, $w_{ik}$ the weight of the connection between the variable i and the neuron k of the hidden layer and $w_k$ is the weight of the connection between the neuron k of the hidden layer and output layer."

In Section 3.1: "For this network, the influence of each input variable on the computed $TCO_2$ is depicted in Fig. S2. The position variables together (latitude, clongitude, slongitude and depth) have no more than 30% influence, allowing biogeochemical variables to be the main ones responsible for the variability of $TCO_2$. Furthermore, the input variable year has an influence lower than 5%. This is probably responsible for capturing the positive interannual trend due to the $TCO_2$ increase derived from anthropogenic emissions of $CO_2$ to the atmosphere (see Sect. 3.2)."

[Figure]

Figure S2. The relative importance of the input variables for NNGv2LDEO. lat: latitude; clon: clongitude; slon: slongitude; temp: temperature; sal: salinity; phosp: phosphate; nit: nitrate; sil: silicate; oxy: oxygen.

RC: Line 254: Why average 1981 to 2015 to obtain 1995 climatology? You have 20 year from 1995 to 2015, but only 14 year from 1981 to 2015.

AR: We averaged 1981 to 2010 because the mean year is 1995. We wrote the wrong year "2015" in the previous manuscript. Therefore, the period has 30 years.

RC: Line 261: This is an important criterion to select the NN for making prediction, but no detail available. Could you supply more information in the supplement material on the influences of position variable of the networks?

AR: Position is essentially added to try to capture processes that change $TCO_2$ that are not reflected through the biogeochemical variables used in this study. For example, the mentioned $A_T$ and $TCO_2$ carried by rivers to the ocean, that modify the typical relations with other variables, like salinity-$A_T$. We have given more information about the influence of the position variable, which was shown in a previous comment.

**RC:** Table 2. Are the errors absolute? If so, please state explicitly. Also, the global errors should be added. Showing negative errors are more meaningful.

**AR**: We have added the bias to Table 2 to show negative errors.

**RC**: Table 6: Does the label "NNGv2" means NNGv2LDEO?

**AR**: It was a mistake. It should be NNGv2LDEO. Thanks. Changed.

**RC**: Table 7: Is the bias absolute? If so, please state explicitly. The global errors should be added. Showing negative errors are more meaningful.

**AR**: Biases were always obtained through the manuscript as the difference between the measured (or computed by other methods) $TCO_2$ and the one computed by the neural network. We have added a clarification in the methods section.

**AC**: "It should be noted that the RMSE and the bias were obtained for all the comparisons, the last statistic being computed as the difference between the measured (or computed by the method to compare) $TCO_2$ and the one obtained with the neural network of the present study."

**RC**: Figure 1b: "y=1x +- 7.8" should be y=1x - 7.8.

**AR**: Thanks. Changed.

**RC**: Figure 3: Showing negative errors are more meaningful.

**AR**: Changed.

**AC**:

[Figure]

Figure 3. Differences between (a) Gv2 TCO2 and NNGv2LDEO TCO2 (0-30 m) and (b) LDEO TCO2 and NNGv2LDEO TCO2 (0 m).

**RC**: Figure 4b: The error bar for depth < 50m should be added using the surface errors.

**AR**: Added.

**AC**:

[Figure]

Figure 4. Histograms and box plots of differences between measured and neural network computed $TCO_2$ in (a) Gv2 and (b) LDEO. *$TCO_2$ computed from measured $pCO_2$ and neural network derived $A_T$.

**RC**: Figure S1.b: There should be a "+" operator between bj*a0 and SUM(wij*ai) in the activation function.

**AR**: Thanks. Changed.

**AC**:

[Figure]

Figure S1. (a) Neural network configuration. The notation is in agreement with Hagan et al. (2014). **a**: input vectors; **W**: weight matrix; **b**: bias matrix; $\sum$: sum; f: transfer function; $\mathbf{a^X}$: output matrix. The superscripts indicate the number of the layer. $cLongitude = cos(\frac{\pi}{180°}longitude)$; $sLongitude = sin(\frac{\pi}{180°}longitude)$. The dimensions of the matrices are for an individual sample. Modified from Hagan et al. (2014). (b) Neuron. $a_i$: inputs to each neuron; $w_{i,j}$: weights of each input to each neuron. Modified from Russell and Norvig et al. (2010).

**RC**: Figure S2: How the std is calculated for T, S, and pCO2. Modelled TCO2 is larger than observed TCO2 for all pCO2 STD > 4. How to explain this?

**AR**: In Fig. S2 the boxplot is obtained by ranges of std of the three mentioned variables: T, S and pCO2. These std were obtained from the monthly average carried out in each 1°x1° pixel from the LDEO data, as is explained in L.160-L.164: "The pCO2, temperature and salinity data from LDEO were monthly-averaged for each year in a 1°x1° grid. The points where the standard deviation (std) of the averaged pCO2, temperature and salinity were greater than ±20 µatm, 1.5°C and 0.5, respectively, were discarded, since the objective is to capture the monthly variability and therefore an extremely high sub-monthly variability could lead to errors."

The measured TCO2 is the larger one, since the boxplot has been obtained with the difference: measured TCO2 minus NN computed TCO2 (as is depicted in the x axis label). With the increase of the std of pCO2, the error increases because of the difficulty to model the high sub-monthly variability of pCO2, which is not the purpose of this study, obtaining a bias of ~2 µatm for pCO2 std > 12.

We have changed the captions in Fig. S2 (now Fig. S3) to clarify how std was calculated.

**AC**:

[Figure]

Figure S3. Box plots of differences between measured and computed $TCO_2$ in LDEO by standard deviation (std) ranges obtained from the monthly average of the LDEO data.

**RC**: Figure S3. If the difference is absolute, please state clearly. Showing negative errors are more meaningful.

**AR**: We have changed Fig. S3 (now Fig. S4) to show negative errors too.

**AC**:

[Figure]

Figure S4. Differences between measured and computed pCO2 with AT from NNGv2 (Broullón et al., 2019) and TCO2 from NNGv2LDEO in LDEO. Units are microatmospheres (µatm).

**RC**: Figure S5 and S6: Showing negative errors are more meaningful.

**AR**: Changed.

**AC**:

[Figure]

Figure S6. Differences between the annual mean of the surface TCO2 neural network climatology and (a) Takahashi et al. (2014) and (b) Lauvset et al. (2016) surface annual mean climatology. Units are micromole per kilogram (μmol kg-1). The color bar was developed in order to show the highest differences beyond the errors of each method. This figure was made with Ocean Data View (Schlitzer, 2016).

[Figure]

Figure S7. Differences between the monthly climatology of TCO2 of Takahashi et al. (2014) and the one of the present study. The color bar was developed in order to show the highest differences beyond the errors of each method. Units are micromole per kilogram (μmol kg$^{-1}$). This figure was made with Ocean Data View (Schlitzer, 2016).

**RC**: Figure S7a: The model produces a much larger seasonal amplitude in the surface water. Unless measurements are not available in all months, the seasonal amplitude of the climatology should be no larger than that of the measurements. Does this indicates either over-fitting or extrapolation in seasons of no measurements.

**AR**: There are no measured values larger than the red limits in BATS (Fig. S7(a); now Fig. S8(a)), but there are in Gv2LDEO dataset, with which the network was trained. Therefore, the larger amplitude is not because overfitting nor extrapolation. Furthermore, the differences are inside the error of the network in the surface layer (Table 2, Central Atlantic: 7.9 μmol/kg (LDEO) and 15.8 μmol/kg (Gv2 in the 0-50 m layer)).

**RC**: Figure S8 and S9. Plotting land with colors confuses grasping the contours of differences.

**AR**: Changed.

**AC**:

[revised manuscript text omitted]

Figure S1. (a) Neural network configuration. The notation is in agreement with Hagan et al. (2014). **a**: input vectors; **W**: weight matrix; **b**: bias matrix; $\sum$: sum; f: transfer function; $\mathbf{a}^X$: output matrix. The superscripts indicate the number of the layer. $cLongitude = cos(\frac{\pi}{180°} longitude)$; $sLongitude = sin(\frac{\pi}{180°} longitude)$. The dimensions of the matrices are for an individual sample. Modified from Hagan et al. (2014). (b) Neuron. $a_i$: inputs to each neuron; $w_{i,j}$: weights of each input to each neuron. Modified from Russell and Norvig et al. (2010).

[Figure]

Figure S2. The relative importance of the input variables for NNGv2LDEO obtained with Eq. (1). lat: latitude; clon: clongitude; slon: slongitude; temp: temperature; sal: salinity; phosp: phosphate; nit: nitrate; sil: silicate; oxy: oxygen.

[Figure]

Figure S3. Box plots of differences between measured and computed TCO$_2$ in LDEO by standard deviation (std) ranges obtained from the monthly average of the LDEO data.

[revised manuscript text omitted]

---

## Author Comment (AC3) · 18 Jun 2020

**RC**: Referee comment; **AR**: author's response; **AC**: author's changes in manuscript.

**Referee #3 response**

**RC**: The authors provide a nice description of global TCO2 distribution based on monthly climatology from a neural network approach. The addition input variable "year" is reasonable and important for TCO2. It is great to see that the neural network outputs match very well with the measured TCO2 for the independent time series locations. This manuscript is well organized and easy to follow. I would like to see the publication of this work.

**AR**: Thank you so much for the thorough revision of the manuscript and for your desire to see this work published. We are pleased to see that you find important to include the year as an input variable of the network to model the positive interannual trend in $TCO_2$ because of the anthropogenic $CO_2$ emissions. We hope to clarify your comments with the answers we give in this document and improve the manuscript. At the end of the answers, the new version of the manuscript is attached for a global view. We have tried to keep a balance to respond appropriately to the 3 reviewers.

**RC**: Lines 71-74, include the influence of mixing on TCO2 variability. For example, upwelling-induced increase in TCO2, which is described in lines 83-86.

**AR**: Added.

**AC**: "Advection and mixing also influence the variability of $TCO_2$ in these two ways (Sabine et al., 2002)."

**RC**: Lines 75-76, the temperature and salinity influence on TCO2 through the modification of the solubility of CO2 affects the seawater pCO2 (which is almost instantaneous) and thus the air-sea CO2 flux, which eventually drives the change in TCO2 over time.

**AR**: Added

**RC**: Line 131, delete de in "minimize de errors"

**AR**: Thanks. Deleted.

**RC**: Lines 142-143, move the full list of the input variables to line 153 to make it ahead of the sentence, "In addition to the target variable…"

**AR**: In L.142-143 we are showing the main changes between the present study and that of Broullón et al. (2019) and the list of the input variables is not a difference between both studies. Therefore, we think the list is more suitable in the original place.

**RC**: Line 147, specify which generalization method was used to prevent overfitting.

**AR**: Added more information.

**AC**: "… the generalization of the network (to prevent overfitting, maintaining a similar error in the training and in the test sets) on the other hand."

**RC**: Line 258, provide how many hidden layers are used in the neural network

**AR**: Added.

**AC**: "The number of neurons tested in the unique hidden layer…"

**RC**: Line 263, the errors cannot be avoided in any case. It is likely that a much smaller weight is assigned to a variable that contains a large error.

**AR**: A well trained neural network can avoid the errors in data, that is, avoiding the overfitting caused by a high number of neurons that makes the network to fit the error in the data. Here is a typical example where the function to fit is a trigonometric function:

[Figure]

The blue line is the function to fit. The black dots are the data with errors used to train the network. The black line is the response of an overfitted neural network which is able to fit all the data with their errors and therefore with a low capacity to reproduce independent data, that is, a bad generalization. This problem comes from training the network with a high number of neurons and/or to maintain the training during a lot of iterations.

**RC**: Line 276, provide how depth-weighted RMSE is calculated.

**AR**: Added to Table 2

**AC**: "For each depth range, the RMSE (bias) in each area was weighted by the contribution of its data to the total."

**RC**: Lines 349-351, the error may come from the bias in total alkalinity. I agree that the error involved in the calculation of pCO2 from AT and TCO2 can be large, but the monthly averaged error is another case. In addition, the choice of K1 and K2 affects the calculated pCO2. Lueker et al. (2000) is a better choice in the case of calculating pCO2. Provide the bias information for total alkalinity in each region in Table 4.

**AR**: Added. We have also added the errors for $TCO_2$.

**AC**:

Table 4. RMSE and bias between measured $pCO_2$ (and in some cases, computed from measured $A_T$ and $TCO_2$ in time series where $pCO_2$ was not measured) and computed $pCO_2$ with $A_T$ from NNGv2 (Broullón et al., 2019) and $TCO_2$ from NNGv2LDEO in several time series. The time period for $pCO_2$ from this study is the same as in Table 3. Consult Table 2 in Landschützer et al. (2014) for its time period. The depth range is 0-15 m. Only time series with more than 30 data points are included. RMSE and bias for computed $A_T$ with NNGv2

**(Broullón et al., 2019) and TCO₂ with NNGv2LDEO are included to show the errors in the variables used to compute TCO₂.**

| Time series | pCO₂ | | | | $A_T$ | | TCO₂ | |
| :---: | :---: | :---: | :---: | :---: | :---: | :---: | :---: | :---: |
| | NNGv2LDEO | | Landschützer et al., 2014 | | NNGv2 (Broullón et al. 2019) | | NNGv2LDEO | |
| | RMSE (µatm) | Bias (µatm) | RMSE (µatm) | Bias (µatm) | RMSE (µatm) | Bias (µatm) | RMSE (µatm) | Bias (µatm) |
| BATS | 17.2 | 9.7 | 15.6 | 0.4 | 5.6 | -1.7 | 10.1 | 4.4 |
| HOT ALOHA SURF | 10.3 | -3.6 | 11.6 | 0.1 | 5.0 | 0.9 | 6.5 | -1.6 |
| ESTOC | 10.6 | 2.7 | 14.5 | -7.1 | 2.6 | -2.7 | 5.3 | -0.6 |
| ICELAND | 16 | 14.8 | - | - | 5.4 | 0.7 | 5.4 | 5.4 |
| IRMINGER | 13.1 | -1.8 | 22.6 | -1.1 | 7.0 | -0.4 | 6.6 | -1.1 |
| K2 | 18.1 | -3.2 | 27.8 | -0.2 | 5.1 | -0.5 | 5.7 | -2.4 |
| KNOT | 20.8 | 8.6 | - | - | 6.6 | -7.3 | 8.2 | -2.5 |

In Sect. 3.2: "Considering the error involved in the calculation of pCO₂ from $A_T$ and TCO₂ (~6 µatm; Millero, 1995) and the error in the computed $A_T$ and TCO₂ with the neural networks (Table 4),…"

**RC**: Line 420-430, Fig. 9, the differences in the surface water at BATS in May, June, July, and August seem large as compared to the values in other months. Provide an explanation.

**AR**: Added explanation.

**AC**: "In the upper 30 m of the water column, the climatology of TCO₂ differs from the measured BATS data from May to August. This difference is mainly explained by the surface error of the network showed for this time series in Fig. 5a, where the computed TCO₂ decreases from maximum to minimum sooner than the measured TCO₂."

**RC**: Line 445-448, the bias for pCO2, considering it is the averaged value over space and time, seems large as compared to the bias in Landschutzer et al. (2017) as shown in Table 4. Therefore, I don't agree that climatologies of other seawater CO2 system variables can be confidently computed. Furthermore, this statement compromises the value of this study if TCO2 can be confidently computed from pCO2 and AT.

**AR**: Bias is large compared to the one in Landschutzer et al. (2014) for 3 of 5 time series stations, but Lanschützer's RMSE is higher in 4 of 5 time series. Regarding the two statistical parameters, the two methods have their strengths and weaknesses. Furthermore, the conclusion in these lines of the manuscript is related to the similarities between our $pCO_2$ climatology and the one of Landschützer et al. (2017) (Figs. S8 and S9), not related to the time series. On the other hand, $TCO_2$ could be confidently computed from $pCO_2$ and $A_T$ but only in the 0 m layer, therefore it should not compromise our study since we compute $TCO_2$ from surface to 5500 m.

**RC**: Fig. 2, color-coded depth instead of the log10 frequency would provide more information regarding the distribution of errors.

**AR**: The high data density, as it is showed in each pixel, would not allow the depths to be displayed properly since data points for different depths are overlapped. We think that the errors with depth can be easily consulted in Table 2.

**RC**: Fig. S4, add another panel with a similar generated from measured bottle TCO2 data. Furthermore, add one more panel to show the differences between the derived TCO2 and the bottle TCO2. The differences are partially shown in Table 2 with the average differences by region. An additional panel regarding the differences would help to determine the reliability of this new dataset on a smaller scale (e.g., 1 degree by 1 degree).

**AR**: It would be interesting to see the differences in the vertical coordinate. However, the comparison would be between data measured in a specific time and climatological data, these differences not reflecting the reliability of the data product. Therefore, we consider Table 2 is a better option to see the differences and Fig. S4 (now Fig. S5) is more adequately to show and describe the distribution pattern of $TCO_2$ in depth.

[revised manuscript text omitted]

Figure S1. (a) Neural network configuration. The notation is in agreement with Hagan et al. (2014). **a**: input vectors; **W**: weight matrix; **b**: bias matrix; $\sum$: sum; f: transfer function; $\mathbf{a^X}$: output matrix. The superscripts indicate the number of the layer. $cLongitude = cos(\frac{\pi}{180°}longitude)$; $sLongitude = sin(\frac{\pi}{180°}longitude)$. The dimensions of the matrices are for an individual sample. Modified from Hagan et al. (2014). (b) Neuron. $a_i$: inputs to each neuron; $w_{i,j}$: weights of each input to each neuron. Modified from Russell and Norvig et al. (2010).

[Figure]

Figure S2. The relative importance of the input variables for NNGv2LDEO obtained with Eq. (1). lat: latitude; clon: clongitude; slon: slongitude; temp: temperature; sal: salinity; phosp: phosphate; nit: nitrate; sil: silicate; oxy: oxygen.

[Figure]

Figure S3. Box plots of differences between measured and computed TCO₂ in LDEO by standard deviation (std) ranges obtained from the monthly average of the LDEO data.

[revised manuscript text omitted]